# SIRT2 and lysine fatty acylation regulate the transforming activity of K-Ras4a

Hui Jing[1†], Xiaoyu Zhang[1†], Stephanie A Wisner[1], Xiao Chen[1], Nicole A Spiegelman[1], Maurine E Linder[2], Hening Lin[1,3*]

[1]Department of Chemistry and Chemical Biology, Cornell University, Ithaca, United States; [2]Department of Molecular Medicine, Cornell University College of Veterinary Medicine, Ithaca, United States; [3]Department of Chemistry and Chemical Biology, Howard Hughes Medical Institute, Cornell University, Ithaca, United States

**Abstract** Ras proteins play vital roles in numerous biological processes and Ras mutations are found in many human tumors. Understanding how Ras proteins are regulated is important for elucidating cell signaling pathways and identifying new targets for treating human diseases. Here we report that one of the K-Ras splice variants, K-Ras4a, is subject to lysine fatty acylation, a previously under-studied protein post-translational modification. Sirtuin 2 (SIRT2), one of the mammalian nicotinamide adenine dinucleotide (NAD)-dependent lysine deacylases, catalyzes the removal of fatty acylation from K-Ras4a. We further demonstrate that SIRT2-mediated lysine defatty-acylation promotes endomembrane localization of K-Ras4a, enhances its interaction with A-Raf, and thus promotes cellular transformation. Our study identifies lysine fatty acylation as a previously unknown regulatory mechanism for the Ras family of GTPases that is distinct from cysteine fatty acylation. These findings highlight the biological significance of lysine fatty acylation and sirtuin-catalyzed protein lysine defatty-acylation.

DOI: https://doi.org/10.7554/eLife.32436.001

*For correspondence:
hl379@cornell.edu

†These authors contributed equally to this work

Competing interests: The authors declare that no competing interests exist.

## Introduction

Protein fatty acylation facilitates direct association of proteins with particular membranes in cells and plays a vital role in protein trafficking, cell signaling, protein-protein interactions, and protein activity (*Lanyon-Hogg et al., 2017*; *Peng et al., 2016*; *Tate et al., 2015*). Dysregulation of protein fatty acylation is implicated in human cancer and neurodegenerative diseases (*Lanyon-Hogg et al., 2017*). While early studies have focused on N-terminal glycine myristoylation and cysteine palmitoylation, little is known about lysine fatty acylation (*Lanyon-Hogg et al., 2017*; *Tate et al., 2015*). Although first reported over two decades ago, the biological function of protein lysine fatty acylation is not clear and to date, only a few proteins, such as tumor necrosis factor α (TNF-α) and interleukin 1-α, are known to be regulated by lysine fatty acylation (*Bursten et al., 1988*; *Hedo et al., 1987*; *Pillai and Baltimore, 1987*; *Stevenson et al., 1993*; *Stevenson et al., 1992*). The enzymes that catalyze the addition or removal of lysine fatty acylation were not known until recently when we and others found that several sirtuins, the nicotinamide adenine dinucleotide (NAD)-dependent protein lysine deacylase, could act as lysine defatty-acylase. We have previously reported that TNF-α (*Jiang et al., 2013*; *Jiang et al., 2016*) and exosome (*Zhang et al., 2016*) secretion are regulated by sirtuin 6 (SIRT6)-catalyzed removal of lysine fatty acylation, demonstrating that lysine fatty acylation is reversible and physiologically important. Other sirtuin family proteins, SIRT1-3 (*Bao et al., 2014*; *Feldman et al., 2013*; *He et al., 2014*; *Liu et al., 2015*; *Teng et al., 2015*) and SIRT7 (*Tong et al., 2017*) have also been found to efficiently remove fatty acyl groups from lysine residues in vitro, suggesting that lysine fatty acylation may be more prevalent. Therefore, we sought to identify other proteins that may be regulated by lysine fatty acylation.

Ras proteins are small GTPases that play important roles in numerous tumor-driving processes, including proliferation, differentiation, survival, cell cycle entry and cytoskeletal dynamics (*Malumbres and Barbacid, 2003*). They act as binary switches: they are active when GTP is bound, turning on specific signaling pathways by recruiting effector proteins, and inactive in the GDP-bound state (*Hancock, 2003*; *Malumbres and Barbacid, 2003*). Guanine nucleotide exchange factors (GEFs) activate Ras by promoting GDP-GTP exchange, whereas GTPase-activating proteins (GAPs) inactivate Ras by promoting intrinsic GTP hydrolysis (*Malumbres and Barbacid, 2003*). In mammals, *HRAS*, *NRAS*, and *KRAS* proto-oncogenes encode four proteins: H-Ras, N-Ras, K-Ras4a, and K-Ras4b. K-Ras4a and K-Ras4b are the two splice variants encoded by the *KRAS* gene. K-Ras4b has attracted most of the attention because it was assumed to be the more abundant and thus the more important K-Ras isoform mutated in human cancers. However, recent studies have revealed that K-Ras4a is widely expressed in many cancer cell lines and its level is similar to that of K-Ras4b in human colorectal tumors (*Tsai et al., 2015*; *Zhao et al., 2015*). A requirement for oncogenic K-Ras4a in lung carcinogenesis has also been demonstrated in mice (*To et al., 2008*). Thus, there is increasing interest in evaluating K-Ras4a as a therapeutic target and in investigating the regulation of K-Ras4a.

Ras proteins exert their functions at cellular membranes, where they interact with distinct effectors and activate downstream signaling (*Hancock, 2003*). Ras proteins typically have two membrane-targeting signals at the C-terminal hypervariable regions (HVRs). All four Ras proteins are modified by cysteine farnesylation on their CaaX motif. H-Ras and N-Ras contain cysteine palmitoylation as the second membrane targeting signal, whereas K-Ras4b uses a polybasic region (PBR) (*Figure 1a*). K-Ras4a possesses a hybrid membrane targeting motif: multiple lysine residues at the C-terminus (similar to K-Ras4b) as well as cysteine palmitoylation (*Figure 1a*) (*Tsai et al., 2015*; *Zhao et al., 2015*). As we set out to identify lysine fatty acylated proteins, the presence of multiple lysine residues in the Ras HVRs caught our attention. If the lysine residues function simply to promote membrane binding by electrostatics, why are there almost invariably lysine but not arginine residues in the HVRs? The prevalence of lysines in Ras HVRs suggests the possibility that lysine residues are post-translationally modified by fatty acylation. Thus, in this study, we set out to investigate whether Ras proteins are regulated by reversible lysine fatty acylation.

## Results

### H-Ras and K-Ras4a contain lysine fatty acylation

To examine whether the lysine residues in the Ras proteins could be fatty acylated, an alkyne-tagged fatty acid analog, Alk14, was used to metabolically label Ras proteins (*Jiang et al., 2013*). As shown in *Figure 1b*, HEK293T cells transiently expressing FLAG-tagged H-Ras, N-Ras, K-Ras4a, or K-Ras4b were treated with Alk14 (50 μM). We ensured that the overexpression levels of different Ras proteins were similar (*Figure 1—figure supplement 1a*). FLAG-tagged Ras proteins were immunoprecipitated and conjugated to rhodamine-azide (Rh-N$_3$) using click chemistry to allow visualization of fatty acylation by in-gel fluorescence. Hydroxylamine (NH$_2$OH) was then used to remove cysteine palmitoylation. Ras-related protein Ral-A (RalA) (*Nishimura and Linder, 2013*) and Syntaxin-6 (STX6) (*Kang et al., 2008*) were included as controls for the efficiency of NH$_2$OH in removing cysteine palmitoylation. Quantification of the fluorescent signal revealed that NH$_2$OH treatment removed over 95% of the fatty acylation from RalA or STX6. However, H-Ras, N-Ras, and K-Ras4a retained 20%, 13% and 47% NH$_2$OH-resistant fatty acylation over total fatty acylation, respectively, whereas K-Ras4b did not show Alk14 labeling either before or after NH$_2$OH treatment (*Figure 1c*). These data suggest that H-Ras, N-Ras, and K-Ras4a might possess non-cysteine fatty acylation.

To determine whether the NH$_2$OH-resistant fatty acylation could be attributed to the lysine residues in the HVRs of H-Ras, N-Ras, and K-Ras4a, we mutated these lysine (K) residues to arginine (R) and examined the fatty acylation of the WT and KR mutants. The H-Ras-3KR (K167/170/185R) mutant (*Figure 1d*), N-Ras-2KR (K169/170R) mutant (*Figure 1—figure supplement 1b*), and the K-Ras4a-3KR (K182/184/185R) mutant but not K-Ras4a-4KR (K169/170/173/176R) mutant (*Figure 1e and f*) all displayed greatly reduced NH$_2$OH-resistant fatty acylation, implying that H-Ras, N-Ras, and K-Ras might be fatty acylated on the lysine residues in their HVRs. Moreover, K-Ras4a-7KR (4KR and 3KR) showed comparable NH$_2$OH-resistant fatty acylation level to the 3KR mutant, suggesting that K182/

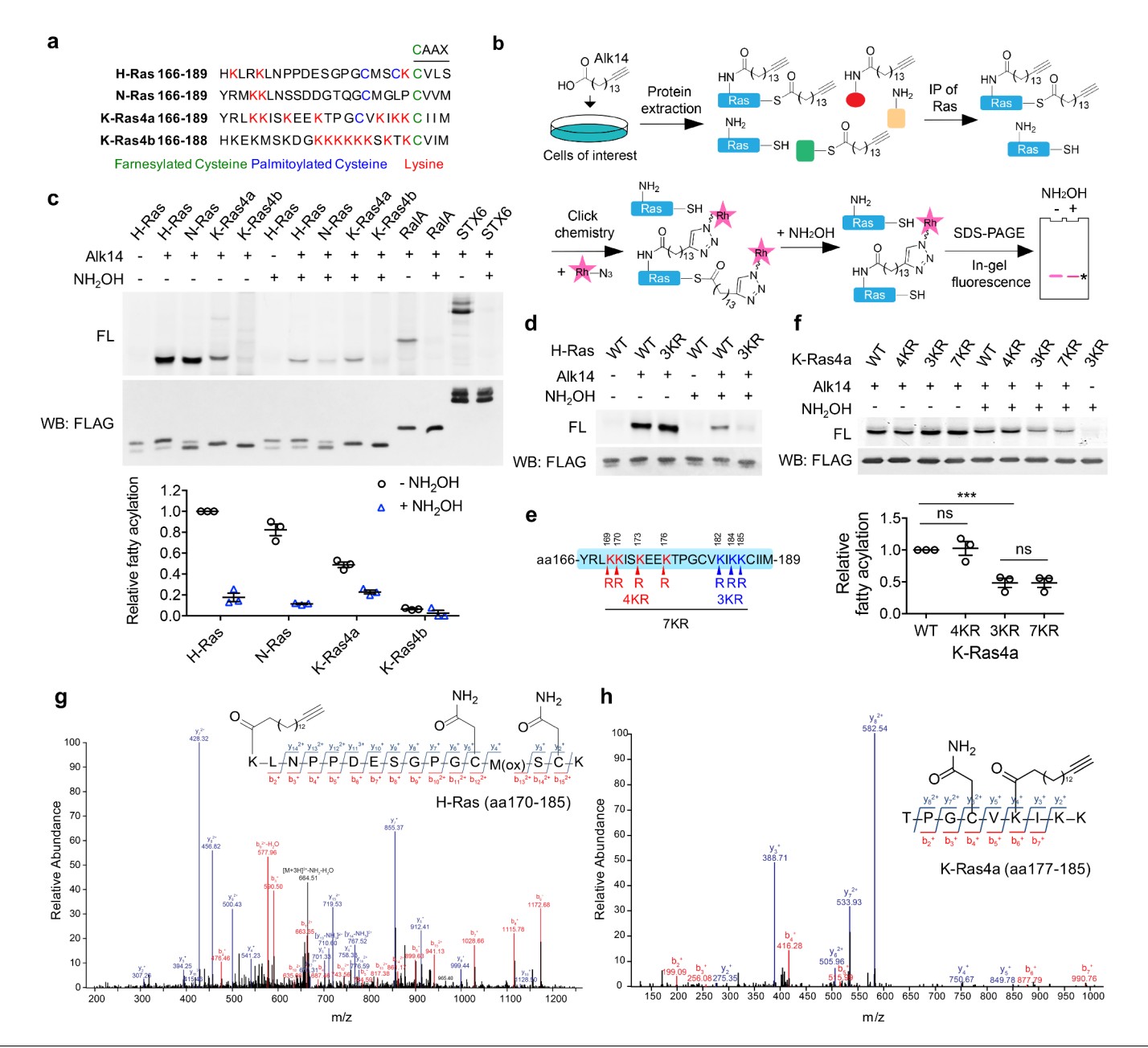

**Figure 1.** H-Ras and K-Ras4a contain lysine fatty acylation. (**a**) Amino acid sequences of the HVRs of Ras proteins. (**b**) Scheme showing the Alk14 metabolic labeling method to study lysine fatty acylation. (**c**) In-gel fluorescence detection of the fatty acylation levels of Ras proteins, RalA and STX6 in HEK293T cells (top panel), and quantification of the relative fatty acylation levels (bottom panel). The fatty acylation level of H-Ras without NH$_2$OH treatment was set to 1. (**d**) In-gel fluorescence showing the fatty acylation levels of H-Ras-WT and -3KR mutant without or with NH$_2$OH treatment. (**e**) Scheme showing the lysine to arginine mutants (4KR, 3KR, and 7KR) used to identify potential fatty acylation sites. (**f**) In-gel fluorescence showing the fatty acylation levels of K-Ras4a-WT and KR mutants without or with NH$_2$OH treatment (top panel) and quantification of fatty acylation levels with NH$_2$OH treatment relative to that of K-Ras4a-WT (bottom panel). (**g, h**) Tandem mass (MS/MS) spectrum of triply charged H-Ras (**g**) and K-Ras4a (**h**) peptides with Alk14-modification on K170 and K182, respectively. The b- and y-ions are shown along with the peptide sequence. The cysteine residues were carbamidomethylated due to iodoacetamide alkylation during sample preparation and methionine was oxidized. FL, fluorescence; WB, western blot. Statistical evaluation was by unpaired two-tailed Student's t test. Error bars represent SEM in three biological replicates. ***p<0.001; ns, not significant. Representative images from three independent experiments are shown.

DOI: https://doi.org/10.7554/eLife.32436.002

The following figure supplement is available for figure 1:

**Figure supplement 1.** N-Ras proteins may be lysine fatty acylated.

*Figure 1 continued on next page*

*Figure 1 continued*

DOI: https://doi.org/10.7554/eLife.32436.003

184/185 of K-Ras4a might be fatty acylated. We then utilized mass spectrometry (MS) to directly identify the lysine fatty acylation of FLAG-H-Ras, -N-Ras or -K-Ras4a extracted from Alk14-treated HEK293T cells with tryptic digestion. This allowed us to identify H-Ras K170 (*Figure 1g*) and K-Ras4a K182 (*Figure 1h*) as being modified, confirming lysine fatty acylation of H-Ras and K-Ras4a. Our attempt to identify N-Ras lysine fatty acylation by MS was not successful possibly because the tryptic peptide with lysine fatty acylation was less abundant (*Figure 1c*), too short (MK$_{acyl}$K) or too hydrophobic (K$_{acyl}$LNSSDDGTQGC$_{cam}$MGLPC$_{prenyl, OMe}$) (*Peng et al., 2016*; *Swaney et al., 2010*; *Tate et al., 2015*).

## SIRT2 catalyzes the removal of lysine fatty acylation from K-Ras4a

Several mammalian sirtuins, including SIRT1, SIRT2, SIRT3, and SIRT6, can efficiently remove fatty acyl groups from protein lysine residues in vitro (SIRT1, SIRT2, and SIRT3) (*Zhang et al., 2017*) and in vivo (SIRT6) (*Bao et al., 2014*; *Feldman et al., 2013*; *Jiang et al., 2013*; *Teng et al., 2015*; *Zhang et al., 2017*). So we next investigated whether any of these sirtuins could remove lysine fatty acylation from H-Ras or K-Ras4a and therefore regulate their function. We incubated H-Ras or K-Ras4a isolated from Alk14-treated HEK293T cells with purified recombinant sirtuins without or with NAD in vitro and examined the H-Ras or K-Ras4a fatty acylation level by in-gel fluorescence after click chemistry. Incubation of H-Ras or K-Ras4a with *Plasmodium falciparum* Sir2A (PfSir2A), a sirtuin family member with robust lysine defatty-acylase activity (*Zhu et al., 2012*), resulted in the removal of most of the NH$_2$OH-resistant fatty acylation from H-Ras and K-Ras4a in the presence of NAD (*Figure 2—figure supplement 1a*). This result further confirmed that the NH$_2$OH-resistant fatty acylation is mainly from lysine residues and indicated that lysine fatty acylation of H-Ras and K-Ras4a is reversible. Furthermore, SIRT2, but not SIRT1, 3, or 6, slightly decreased the lysine fatty acylation signal of H-Ras (*Figure 2a*); SIRT1 and SIRT2, but not SIRT3 and SIRT6, removed lysine fatty acylation from K-Ras4a. Notably, SIRT2 showed better activity than SIRT1 on K-Ras4a lysine fatty acylation (*Figure 2b*). In contrast, SIRT1 and SIRT2 showed little effect on the fatty acylation of K-Ras4a-3KR (*Figure 2—figure supplement 1b*), which exhibited significantly lower lysine fatty acylation than K-Ras4a-WT (*Figure 1f*), suggesting that SIRT1 and SIRT2 do not possess cysteine defatty-acylase activity. An HPLC-based in vitro activity assay also revealed that SIRT2 was unable to remove the cysteine myristoyl group from a K-Ras4a-C180myr peptide (*Figure 2—figure supplement 1c*). Furthermore, knockdown (KD) of SIRT2 in HEK293T cells did not affect lysine fatty acylation of H-Ras (*Figure 2c and f*), whereas KD of SIRT2 but not SIRT1 significantly increased lysine fatty acylation of K-Ras4a compared with control (Ctrl) KD (*Figure 2d,e and f*). We also noted that SIRT2 KD did not affect fatty acylation of N-Ras (*Figure 2—figure supplement 1d*). Taken together, these results illustrate that K-Ras4a is a lysine defatty-acylation substrate for SIRT2 in cells, but H-Ras and N-Ras are not.

We next further validated that K-Ras4a is regulated by SIRT2-mediated defatty-acylation. We utilized the SIRT2-H187Y (HY) mutant, which has previously been shown to be catalytically dead in lysine deacetylation (*North and Verdin, 2007b*), as a negative control. An HPLC-based in vitro assay demonstrated that the H187Y mutation dramatically decreased SIRT2 defatty-acylation activity, while it completely abolished its deacetylation activity (*Figure 2—figure supplement 1e*). Co-expression of SIRT2 with K-Ras4a in HEK293T cells substantially decreased K-Ras4a lysine fatty acylation, whereas co-expression of SIRT2-HY had much less effect (*Figure 2g*), suggesting that K-Ras4a defatty-acylation requires SIRT2 catalytic activity. Interestingly, our finding that mutation of the catalytic histidine residue did not completely abolish sirtuin enzymatic activity is not without precedent. For example, mutating the catalytic histidine of bacterial Sir2Tm (*Hoff et al., 2006*), yeast HST2 (*Jackson et al., 2003*), and human SIRT6 (*Zhang et al., 2016*) also retained some catalytic activity.

To investigate whether K-Ras4a could also be regulated by SIRT2 through deacetylation, we examined its acetylation level using a pan-specific acetyl lysine antibody. Acetylation was not detected on K-Ras4a in either Ctrl KD or SIRT2 KD cells without or with histone deacetylases (HDAC) inhibitor Trichostatin A (TSA) (*Figure 2—figure supplement 1f*). We also searched our K-Ras4a MS

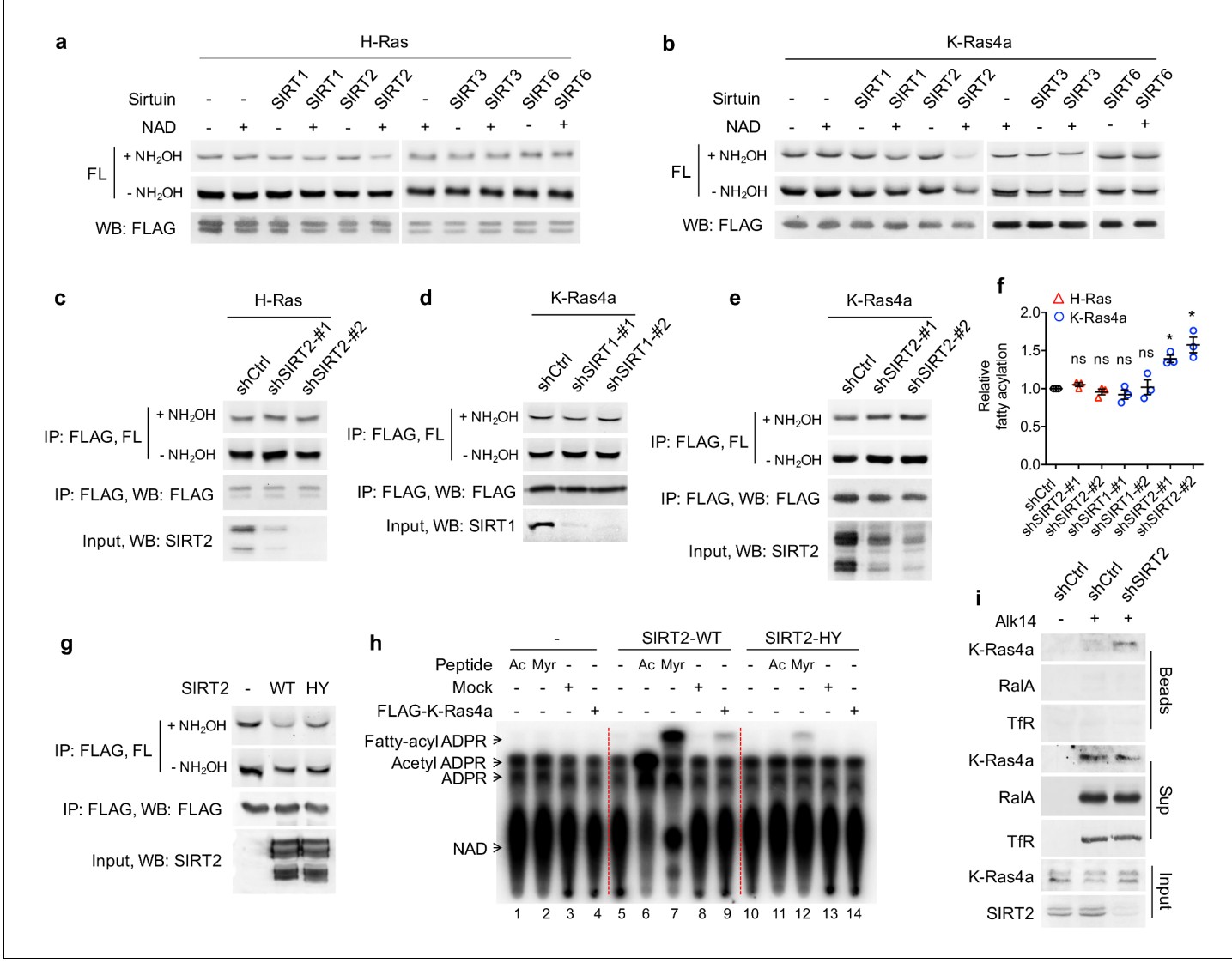

**Figure 2.** SIRT2 removes lysine fatty acylation from K-Ras4a. (**a, b**) In-gel fluorescence detection of fatty acylation of H-Ras (**a**) and K-Ras4a (**b**) treated with 5 µM of SIRT1, 2, 3 and 6 without or with 1 mM of NAD in vitro. (**c**) Effect of SIRT2 KD on the fatty acylation level of H-Ras in HEK293T cells. (**d, e**) Effect of SIRT1 KD (**d**) and SIRT2 KD (**e**) on the fatty acylation level of K-Ras4a in HEK293T cells. (**f**) Quantification of the fatty acylation levels with $NH_2OH$ treatment in (**c**), (**d**) and (**e**). The fatty acylation level in the corresponding Ctrl KD was set to 1. (**g**) Effect of overexpressing SIRT2-WT and SIRT2-HY catalytic mutant on K-Ras4a fatty acylation level. (**h**) Fatty acylated lysine in K-Ras4a detected by formation of $^{32}$P-labeled fatty acyl-ADPR using $^{32}$P-NAD. (**i**) Lysine fatty acylation of endogenous K-Ras4a in Ctrl and SIRT2 KD (by shSIRT2-#2) HCT116 cells detected by Alk14 labeling and biotin pull-down. Sup, supernatant; Ac, H3K9 acetyl peptide; Myr, H3K9 myristoyl peptide. Statistical evaluation was by unpaired two-tailed Student's t test. Error bars represent SEM in three biological replicates. *p<0.05; ns, not significant. Representative images from three independent experiments are shown.

DOI: https://doi.org/10.7554/eLife.32436.004

The following figure supplements are available for figure 2:

**Figure supplement 1.** SIRT2 regulates lysine fatty acylation of K-Ras4a.

DOI: https://doi.org/10.7554/eLife.32436.005

**Figure supplement 2.** Endogenous K-Ras4a is lysine fatty acylated.

DOI: https://doi.org/10.7554/eLife.32436.006

data and did not find any peptides with lysine acetylation, indicating that SIRT2 likely does not regulate K-Ras4a via deacetylation.

With SIRT2 as a tool, we further confirmed the existence of lysine fatty acylation on K-Ras4a in cells that were not treated with Alk14. We used a previously developed assay that relies on $^{32}$P-NAD to detect sirtuin-catalyzed deacylation reactions (*Du et al., 2011*). When histone H3K9 acetyl (Ac) and myristoyl (Myr) peptides were incubated with SIRT2-WT in the presence of $^{32}$P-NAD, the formation of the acyl-ADPR product could be detected by autoradiography after separation using thin-layer chromatography (TLC) (*Figure 2h*, lanes 6 and 7). In contrast, the SIRT2-HY mutant only generated a tiny amount of the acyl-ADPR product (*Figure 2h*, lanes 11 and 12). When K-Ras4a isolated from HEK293T cells was treated with SIRT2-WT in the presence of $^{32}$P-NAD, a spot corresponding to fatty acyl-ADPR but not acetyl-ADPR was detected (*Figure 2h*, lane 9). Control reactions without SIRT2 (*Figure 2h*, lane 4) or with the HY mutant (*Figure 2h*, lane 14) did not generate the fatty acyl-ADPR product. These results demonstrate that K-Ras4a contains lysine fatty acylation that can be removed by SIRT2 in the absence of Alk14 supplementation. A peptide carrying palmitoylation, but not myristoylation, on K182 of FLAG-tagged K-Ras4a was detected by MS (*Figure 2—figure supplement 1g*), demonstrating that palmitoylation is the major native lysine acylation of K-Ras4a.

We then investigated whether endogenous K-Ras4a is also regulated by SIRT2-catalyzed lysine defatty-acylation. For this purpose, we used the HCT116 human colorectal cancer cell line, in which K-Ras4a was shown to be expressed (*Tsai et al., 2015*). Since the commercial antibody against K-Ras4a did not immunoprecipitate K-Ras4a, we enriched fatty acylated proteins labeled with Alk14 as previously described (*Wilson et al., 2011*), and detected fatty acylated K-Ras4a via Western blot using a K-Ras4a-specific antibody. HCT116 cells with Ctrl or SIRT2 KD were cultured in the presence of Alk14. Proteins were then extracted and a biotin affinity tag was attached to the Alk14-labeled proteins with click chemistry. The biotin-conjugated proteins were pulled down using streptavidin beads, and subsequently washed with 1% SDS to disrupt protein-protein interaction. Proteins that were only fatty acylated on cysteine residues were then released from the streptavidin beads into the supernatant (Sup) *via* NH$_2$OH treatment, while proteins with lysine fatty acylation were retained. As shown in *Figure 2i*, RalA (*Figure 1c*) and transferrin receptor (TfR) (*Omary and Trowbridge, 1981*), which are predominantly cysteine fatty acylated, were present in the supernatant but barely detectable from the streptavidin beads, indicating that the NH$_2$OH treatment was effective. In Ctrl KD cells, K-Ras4a was mainly detected in the supernatant. However, in the SIRT2 KD cells, K-Ras4a was detected both on the streptavidin beads and in the supernatant, indicating that endogenous K-Ras4a possesses lysine fatty acylation that is regulated by SIRT2.

By immunoprecipitation of total Ras protein from Alk14-treated HCT116 cells using a pan-Ras (Y13-259) antibody, we found that endogenous Ras proteins exhibited NH$_2$OH-resistant fatty acylation (*Figure 2—figure supplement 2a*). Moreover, SIRT2 KD increased the NH$_2$OH-resistant fatty acylation of Ras proteins (*Figure 2—figure supplement 2b*). Since SIRT2 KD did not affect lysine fatty acylation of overexpressed H-Ras (*Figure 2c and f*) and N-Ras (*Figure 2—figure supplement 1d*), the data suggested that the increase in total Ras lysine fatty acylation observed in SIRT2 KD cells can be attributed to K-Ras4a lysine fatty acylation. This result, together with the detection of K-Ras4a lysine fatty acylation by Alk14 biotinylation, further supports that endogenous K-Ras4a is lysine fatty acylated and is regulated by SIRT2-mediated lysine defatty-acylation. We also performed MS analysis of endogenous Ras immunoprecipitated from HCT116 cells treated with SIRT2 shRNA and Alk14. We identified a peptide with a primary mass matching the Alk14-modified K-Ras4a aa177-185 peptide, whose exact m/z and isotope pattern were the same as those of overexpressed K-Ras4a (*Figure 2—figure supplement 2c*). However, this primary mass did not trigger MS2, which was likely due to low peptide abundance (*Figure 2—figure supplement 2c and d*). It has been shown that K-Ras has a much lower expression level than H-Ras and N-Ras because of its rare codon bias (*Ali et al., 2017*; *Lampson et al., 2013*).

SIRT2 was reported to reside predominately in the cytoplasm (*Inoue et al., 2007*; *North and Verdin, 2007a*). The regulation of K-Ras4a lysine fatty acylation by SIRT2 suggested that SIRT2 might also exist at cellular membranes, where K-Ras4a mainly resides. Indeed, by subcellular fractionation, we found that SIRT2 was present in both soluble and membrane fractions (*Figure 2—figure supplement 1h*). Co-immunoprecipitation (co-IP) revealed K-Ras4a associated with endogenous SIRT2 (*Figure 2—figure supplement 1i*). These results further support that K-Ras4a is a lysine defatty-acylase substrate for SIRT2.

# Mapping the fatty acylated lysine residues regulated by SIRT2

MS results suggested that K182 was the preferentially fatty acylated lysine on K-Ras4a. However, the lysine 182 to arginine mutant (K182R) exhibited similar lysine fatty acylation levels to that of WT (*Figure 3a*). As the 3KR (K182/184/185R) but not the 4KR (K169/170/173/176R) mutant significantly decreased K-Ras4a lysine fatty acylation (*Figure 1f*), we also mutated K184 and K185 to arginine individually. Neither the K184R nor K185R mutation decreased lysine fatty acylation as the 3KR mutant did (*Figure 3a*). These results suggested that K182, 184 and 185 were likely to be modified redundantly. We suspected that it was hard to pinpoint the exact modification site by mutagenesis because the K182R mutation might enhance fatty acylation on the other two nearby lysine residues. To test this hypothesis, we performed MS analysis of FLAG-K-Ras4a-K182A extracted from Alk14-treated HEK293T cells. We tested the K182A instead of K182R mutant because the K182R mutant would produce a tryptic peptide that is too short to be detected. As expected, the K182A mutation

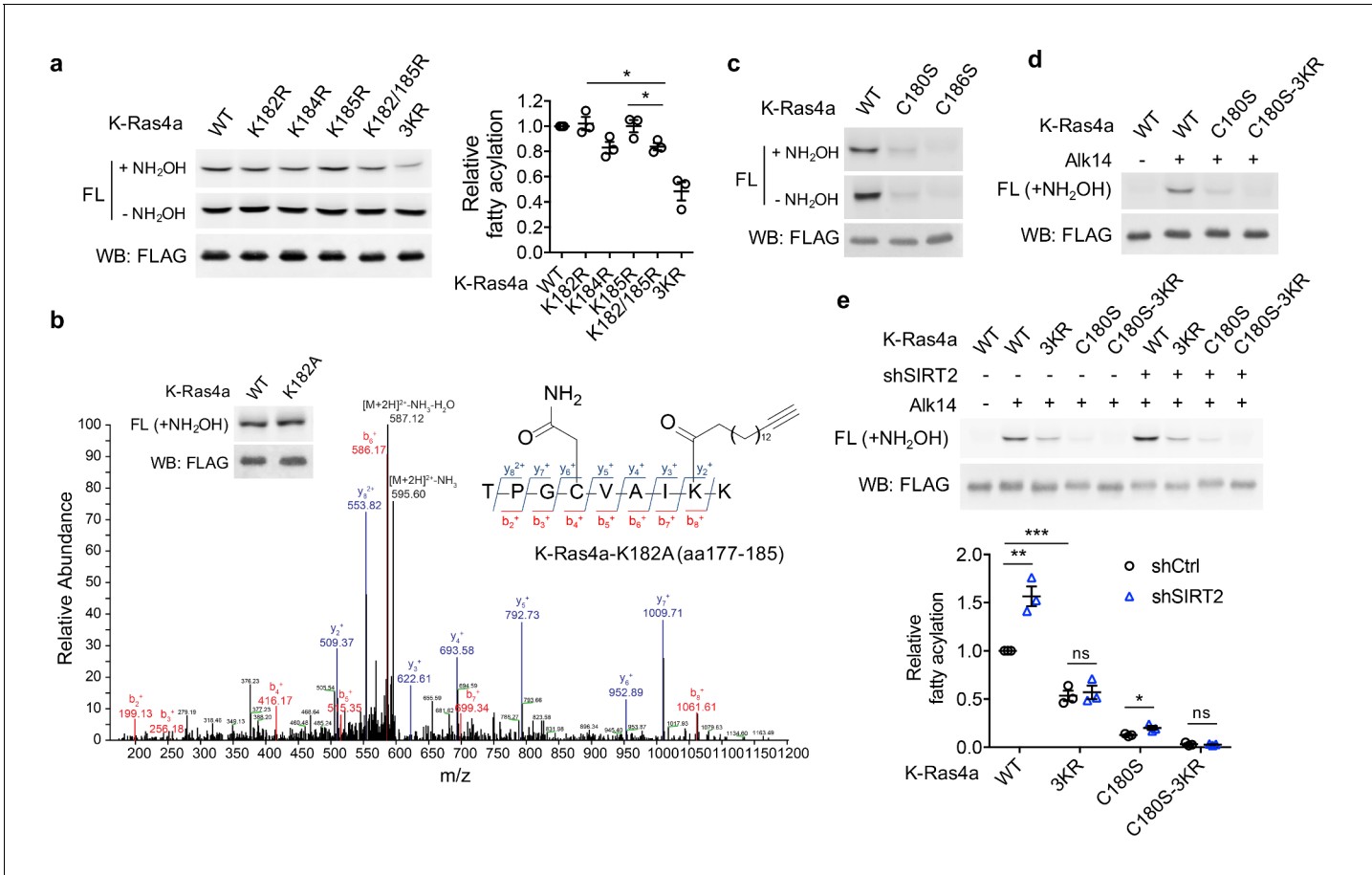

**Figure 3.** SIRT2 regulates lysine fatty acylation of K-Ras4a on K182/184/185. (**a**) Fatty acylation levels of K-Ras4a-WT, -K182R, -K184R, -K185R, -K182/185R, and -3KR by in-gel fluorescence (left panel) and quantification of fatty acylation levels after $NH_2OH$ treatment relative to that of K-Ras4a-WT (right panel). (**b**) MS/MS spectrum of triply charged K-Ras4a-K182A peptide with Alk14 modification on K184. The b- and y-ions are shown along with the peptide sequence. The cysteine residue was carbamidomethylated due to iodoacetamide alkylation during sample preparation. Fatty acylation levels of K-Ras4a-WT and -K182A with $NH_2OH$ were also shown. (**c**) Fatty acylation levels of K-Ras4a-WT, -C180S, and -C186S. (**d**) Fatty acylation levels of K-Ras4a-WT, -C180S and -C180S-3KR after $NH_2OH$ treatment. (**e**) Fatty acylation levels of K-Ras4a-WT, -3KR, -C180S and -C180S-3KR after $NH_2OH$ treatment in Ctrl and SIRT2 KD (by shSIRT2-#2) HEK293T cells. Quantification of the fluorescent intensity relative to K-Ras4a-WT is shown in the bottom panel. Statistical evaluation was by unpaired two-tailed Student's t test. Error bars represent SEM in three biological replicates. *p<0.05; **p<0.01; ***p<0.001. Representative images from three independent experiments are shown.

DOI: https://doi.org/10.7554/eLife.32436.007

The following figure supplement is available for figure 3:

**Figure supplement 1.** SIRT2 removes lysine fatty acylation from K-Ras4a-WT, —4KR and -C180S, but not —3KR.

DOI: https://doi.org/10.7554/eLife.32436.008

did not affect overall level of K-Ras4a lysine fatty acylation (*Figure 3b*). A peptide (amino acids 177–185) fatty acylated on K184 was detected by MS (*Figure 3b*), which agrees with our hypothesis. It was likely that K185 could also be fatty acylated for the K182R mutant, because the K182/185R mutant slightly but significantly decreased lysine fatty acylation levels compared with the K182R and K185R single mutants (*Figure 3a*). The K185 fatty acylation was not detected by MS most likely because the modified tryptic peptide was too short and hydrophobic ($K_{fatty-acyl}C_{prenyl, \, oMe}$). Overall, these data indicate that K182/184/185 are fatty acylated redundantly and that the 3KR mutation is needed to abolish lysine fatty acylation on the C-terminus of K-Ras4a.

K-Ras4a has been shown to be prenylated on cysteine 186 and palmitoylated on cysteine 180 (*Tsai et al., 2015*). To examine whether cysteine prenylation or palmitoylation plays a role in lysine fatty acylation, we generated cysteine-to-serine C180S and C186S mutants. Mutation of the prenyl-cysteine (C186S) completely abolished the fatty acylation of K-Ras4a (*Figure 3c*), which is consistent with the model that prenylation of the cysteine on the CaaX motif of Ras proteins is required for the subsequent fatty acylation (*Wright and Philips, 2006*). On the other hand, mutation of the palmitoy-lated cysteine (C180S) led to a substantial, but not a complete, loss of K-Ras4a lysine fatty acylation (*Figure 3c*). The fatty acylation on the C180S mutant was $NH_2OH$-resistant and was abolished by combining the C180S and 3KR mutations (*Figure 3d*), implying that the C180S mutant was fatty acyl-ated on K182/184/185. These data suggest that cysteine palmitoylation might play an important but nonessential role in the occurrence of lysine fatty acylation. It is possible that cysteine palmitoylation facilitates the lysine fatty acyl transfer reaction, or the delivery of K-Ras4a to where lysine fatty acyla-tion occurs.

We next assessed whether SIRT2 regulates fatty acylation of K-Ras4a on K182/184/185. SIRT2 removed lysine fatty acylation from K-Ras4a-WT, the 4KR mutant, and the C180S mutant, but not the 3KR mutant in vitro (*Figure 3—figure supplement 1a and b*). SIRT2 KD in HEK293T cells increased lysine fatty acylation of K-Ras4a-WT and the C180S mutant, but not the 3KR and C180S-3KR mutants (*Figure 3e*). A similar result for K-Ras4a-WT and the-3KR mutant was also observed in *Sirt2* wildtype (WT) and knockout (KO) mouse embryonic fibroblast (MEF) cells (*Figure 3—figure supplement 1c and d*). These data indicate that fatty acylation on K182/184/185 is regulated by SIRT2.

## Lysine fatty acylation regulates subcellular localization of K-Ras4a

We next set out to study the effect of lysine fatty acylation on K-Ras4a. A variety of PTMs on Ras proteins, such as cysteine palmitoylation (*Choy et al., 1999*; *Rocks et al., 2005*), phosphorylation (*Ballester et al., 1987*; *Bivona et al., 2006*), and ubiquitination (*Jura et al., 2006*), function to deliver the molecule to the right place within the cell. We hypothesized that lysine fatty acylation may also be critical for the correct subcellular distribution of K-Ras4a. To test this hypothesis, we fused *Aequorea coerulescens* Green Fluorescent Protein (GFP) to the N-terminus of K-Ras4a-WT and the 3KR mutant and performed live imaging with confocal microscopy in Ctrl and SIRT2 KD HEK293T cells to visualize K-Ras4a localization. The levels of over-expressed K-Ras4a-WT and -3KR were equal in Ctrl and SIRT2 KD cells (*Figure 4—figure supplement 1a*). We also imaged cells with similar GFP intensity under the same settings to avoid potential false positive observations caused by different levels of expression. In Ctrl KD cells, both K-Ras4a-WT and the 3KR mutant displayed predominant localization to the plasma membrane (PM). However, the presence of 3KR on intracellu-lar puncta was noticeably more pronounced compared to WT. SIRT2 KD decreased the intracellular punctate-localized K-Ras4a-WT compared to Ctrl KD, whereas it had no effect on the punctate local-ization of the 3KR mutant (*Figure 4a and b*) that did not contain SIRT2-regulated lysine fatty acyla-tion. This result suggests that SIRT2-catalyzed lysine defatty-acylation promotes the intracellular punctate localization of K-Ras4a and that the K-Ras4a-3KR mutant has increased intracellular punc-tate localization mainly due to the lack of lysine fatty acylation. Consistently, a similar trend was observed in *Sirt2* WT and KO MEF cells (*Figure 4—figure supplement 1b*). Moreover, increased intracellular punctate localization of the 3KR mutation were also observed for K-Ras4a-WT and onco-genic K-Ras4a-G12V (K-Ras4a-G12V exhibited a lysine fatty acylation level comparable to K-Ras4a-WT, *Figure 4—figure supplement 1c*) in HCT116 cells and NIH3T3 cells (*Figure 4—figure supple-ment 1d and e*). On the other hand, SIRT2 KD did not affect the intracellular punctate localization of H-Ras (*Figure 4a and b*), which is consistent with our observation that H-Ras was not regulated by SIRT2 through lysine defatty-acylation. Taken together, these data indicate that lysine fatty acylation

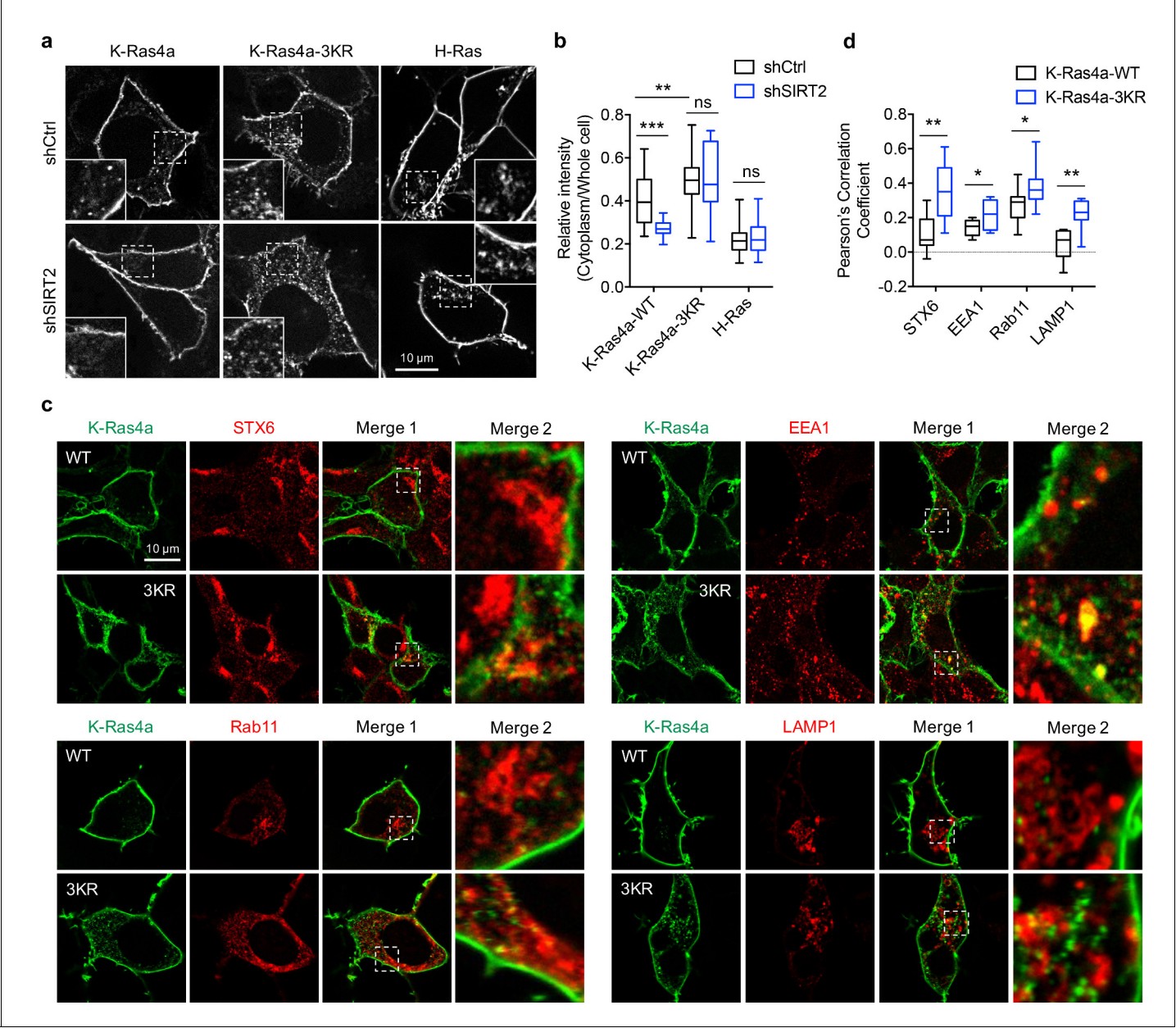

**Figure 4.** Lysine fatty acylation regulates subcellular localization of K-Ras4a. (a) Confocal images showing subcellular localization of GFP-K-Ras4a-WT, -3KR, and GFP-H-Ras in HEK293T cells with Ctrl or SIRT2 KD (by shSIRT2-#2). Insets are magnifications of the regions enclosed by the white dashed squares. (b) Statistical analyses of the relative cytoplasm to whole cell intensity of K-Ras4a-WT, -3KR, and H-Ras from (a) (n = 16, 16, 16, 16, 21, 21 for each sample from left to right, respectively). (c) Images showing the colocalization of GFP-K-Ras4a-WT or-3KR with STX6, EEA1, Rab11, and LAMP1 in HEK293T cells. Merge 2 shows the magnified white dashed squares-enclosed regions in Merge 1. (d) Statistical analyses of the cytoplasmic colocalization of K-Ras4a-WT or -3KR with the indicated intracellular membrane markers from (c) using Pearson's coefficient (n = 11, 11, 11, 11, 17, 17, 10, 10 cells for each sample from left to right, respectively). Statistical evaluation was by two-way ANOVA. *Centre line* of the box plot represents the mean value, box represents the 95% confidence interval, and whiskers represent the range of the values. *$p<0.05$; **$p<0.01$; ***$p<0.001$; ns, not significant. Representative images are shown.

DOI: https://doi.org/10.7554/eLife.32436.009

The following figure supplement is available for figure 4:

**Figure supplement 1.** Lysine fatty acylation regulates subcellular localization of K-Ras4a.

DOI: https://doi.org/10.7554/eLife.32436.010

inhibits the intracellular punctate localization of K-Ras4a and SIRT2 promotes this localization by defatty-acylation. In addition, the K-Ras4a-C180S mutant that lacks cysteine palmitoylation and the majority of lysine fatty acylation extensively localized to internal membranes, which was distinct from the punctate localization of the 3KR mutant that is deficient in lysine fatty acylation but retains cysteine palmitoylation (*Figure 4—figure supplement 1f and g*, *Videos 1*, *2* and *3*). This implies that cysteine palmitoylation might facilitate the punctate localization of K-Ras4a in the absence of lysine fatty acylation, while lysine fatty acylation inhibits it.

It has been shown that Ras proteins associate with and signal from endomembrane compartments, including the endoplasmic reticulum (ER), Golgi, endosomes, and lysosome (*Apolloni et al., 2000*; *Chiu et al., 2002*; *Fivaz and Meyer, 2005*; *Hancock, 2003*; *Howe et al., 2001*; *Jura et al., 2006*; *Lu et al., 2009*; *Misaki et al., 2010*). Therefore, we next set out to identify the endomembrane compartments where lysine defatty-acylated K-Ras4a is localized. We performed colocalization analyses with a series of membrane compartment markers. Compared with K-Ras4a-WT, the 3KR mutant exhibited more pronounced cytoplasmic colocalization with *trans*-Golgi network (TGN) marker STX6, early endosome marker EEA1, recycling endosome marker Rab11, and lysosome marker LAMP1 (*Figure 4c and d*), but not with the ER marker Sec61, *trans*-Golgi marker GalT and late endosome marker Rab7 (*Figure 4—figure supplement 1h and i*). Time-lapse confocal imaging also revealed that K-Ras4a-3KR displayed more internalization from the plasma membrane into punctate structures than did the WT (*Video 1* and *2*). These results suggest that removal of lysine fatty acylation from K-Ras4a promotes its localization to endomembranes in endocytic pathways, by which it may be routed from early endosome to the lysosome for degradation and to the TGN or recycling endosomes to return to the plasma membrane (*Grant and Donaldson, 2009*).

## Lysine fatty acylation regulates transforming activity of K-Ras4a

We next investigated whether lysine fatty acylation also affects the function of K-Ras4a. We assessed the ability of constitutively active K-Ras4a-G12V and the K-Ras4a-G12V-3KR mutant to enable anchorage-independent growth, promote proliferation in monolayer cultures, and stimulate migration in Ctrl and Sirt2 KD NIH3T3 cells. In Ctrl KD cells, expression of K-Ras4a-G12V-3KR resulted in significantly more colony formation on soft agar than did expression of K-Ras4a-G12V. Furthermore, Sirt2 KD caused a greater decrease in colony formation induced by K-Ras4a-G12V (75% decrease) than by K-Ras4a-G12V-3KR (45% decrease) (*Figure 5*). Additionally, Sirt2 KD more potently inhibited

K-Ras4a-G12V-mediated colony formation than H-Ras4a-G12V-mediated colony formation (*Figure 5*), consistent with the fact that SIRT2 regulates lysine fatty acylation of K-Ras4a but not H-Ras or K-Ras4a-3KR. Thus, lysine fatty acylation inhibits the ability of K-Ras4a-G12V to induce anchorage-independent growth of cells and SIRT2 promotes it through defatty-acylation. One caveat of the result, however, was that Sirt2 KD still decreased the colony formation induced by K-Ras4a-G12V-3KR or H-Ras-G12V, whose lysine fatty acylation was not regulated by SIRT2. This is not unexpected because SIRT2 is known to exert tumor-promoting functions by deacetylating various targets (*He et al., 2012*; *Hu et al., 2014*; *Jing et al., 2016*; *Jing and Lin, 2015*; *Liu et al., 2013*; *Wang et al., 2014*; *Xu et al., 2016*; *Zhao et al., 2013*; *Zhou et al., 2016*). Thus, the effect of Sirt2 KD on K-Ras4a-G12V-3KR- and H-Ras-induced transformation might be attributed to other substrates for SIRT2.

In monolayer cultures, NIH3T3 cells expressing K-Ras4a-G12V-3KR displayed a higher proliferation rate than those expressing K-Ras4a-G12V. Sirt2 KD inhibited the proliferation of the

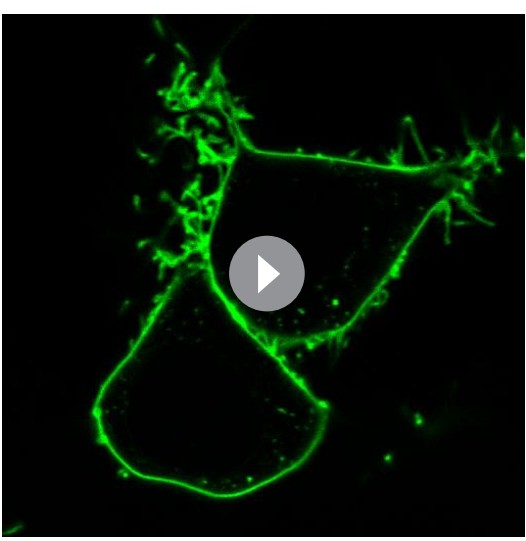

**Video 1.** Dynamics of K-Ras4a in HEK293T cells. Cells were transfected with GFP-K-Ras4a for 24 hr before being subjected to time-lapse confocal microscopy. Images were collected at 1 s intervals for 1 min of a single plane.

DOI: https://doi.org/10.7554/eLife.32436.011

NIH3T3-K-Ras4a-G12V cells (47% inhibition) slightly more than that of the NIH3T3-K-Ras4a-G12V-3KR (34% inhibition) cells (*Figure 5—figure supplement 1a*). Thus, lysine fatty acylation negatively regulates K-Ras4a-G12V-induced cell proliferation under monolayer culture conditions, but the effect was smaller than that on anchorage-independent growth (*Figure 5*). Results from transwell migration assays revealed that the 3KR mutation did not affect the capability of K-Ras4a-G12V to induce cell migration. Consistent with this finding, Sirt2 KD decreased K-Ras4a-G12V and K-Ras4a-G12V-3KR-mediated cell migration similarly (*Figure 5—figure supplement 1b and c*). Therefore, lysine fatty acylation does not affect the ability of K-Ras4a-G12V to stimulate cell migration.

## Lysine fatty acylation of K-Ras4a affects interaction with A-Raf

The dynamic regulation of Ras localization is known to be closely coupled to its signaling output (*Chandra et al., 2011*; *Chiu et al., 2002*; *Dekker et al., 2010*). We decided to further explore the molecular mechanism underlying the regulation of K-Ras4a-mediated transformation by lysine fatty acylation. We first sought to examine whether lysine fatty acylation affects K-Ras4a activation by a pull-down assay with the Ras-binding domain (RBD) of Raf1, which only binds to the GTP-bound form of Ras (*Chiu et al., 2002*). Neither the 3KR mutation nor SIRT2 KD affected EGF-stimulated GTP loading of K-Ras4a (*Figure 6—figure supplement 1a and b*) or the constitutively GTP-loaded state of K-Ras4a-G12V (*Figure 6—figure supplement 1c and d*). We then determined whether K-Ras4a at endomembranes exists in its GTP-bound state using DsRed-RBD (DsRed fused to the N-terminus of RBD) as a probe. Notably, we observed more colocalization of DsRed-RBD with K-Ras4a on intracellular puncta in cells expressing the 3KR mutant (*Figure 6—figure supplement 1e,f,g*) than in cells expressing K-Ras4a-WT. Furthermore, SIRT2 KD decreased the colocalization of DsRed-RBD with K-Ras4a-WT at intracellular puncta, but not with the 3KR mutant (*Figure 6—figure supplement 1e*).

The results above suggest that SIRT2-dependent lysine defatty-acylation may promote the localization of activated (GTP-loaded) K-Ras4a at endomembranes, which raises the possibility that lysine defatty-acylation may alter the signaling specificity of K-Ras4a by recruiting different effector proteins to endomembranes. We therefore investigated whether lysine defatty-acylation influenced the binding and activation of the three most well characterized Ras effectors: Raf1, PI3K, and RalGDS (*Berndt et al., 2011*). Co-immunoprecipitation demonstrated that neither the 3KR mutation nor Sirt2 KD altered the binding of K-Ras4a-G12V with Raf1, PI3K, or RalGDS (*Figure 6—figure supplement 2a*). We also assessed the capacity of K-Ras4a-G12V and -G12V-3KR in Ctrl and Sirt2 KD cells to activate Raf1, PI3K, and RalGDS signaling pathways using phosphorylated Erk, phosphorylated Akt, and phosphorylated Jnk as reporters, respectively. K-Ras4a-G12V and -G12V-3KR induced comparable levels of Erk activation, which was not affected by Sirt2 KD. Sirt2 KD resulted in a reduction of Akt and Jnk activation, but the effect was similar for both K-Ras4a-G12V and -G12V-3KR (*Figure 6—figure supplement 2b*), suggesting that other Sirt2 targets are important for Akt and Jnk activation. These results suggest that SIRT2 catalyzed lysine defatty-acylation of K-Ras4a does not affect the activation of Raf1, PI3K or RalGDS by K-Ras4a.

To identify proteins whose binding to K-Ras4a is regulated by lysine fatty acylation, we performed a protein interactome study using stable isotope labeling by amino acids in cell culture (SILAC) (*Figure 6—figure supplement 3a*). We cultured NIH3T3 cells with stable K-Ras4a-G12V and K-Ras4a-G12V-3KR overexpression in light-isotope- and heavy-isotope-labeled media, respectively. We then performed FLAG IP, mixed the eluted fractions from both IPs, digested with trypsin and analyzed by MS to identify proteins with Heavy/Light (H/L) ratios > 1.3 or < 0.77, which were candidates that would potentially bind to K-Ras4a-G12V and K-Ras4a-G12V-3KR differently. The experiment was also repeated after swapping the heavy and light SILAC labels. Additionally, to confirm that the effect of the 3KR mutation on the K-Ras4a-G12V interactome was due to the lack of lysine fatty acylation, we also examined the K-Ras4a-G12V interactome in Ctrl and Sirt2 KD cells with SILAC, which enabled the identification of proteins (H/L > 1.3 or<0.77) whose binding to K-Ras4a-G12V was regulated by SIRT2. Integration of the three interactome experiments resulted in 175 interacting proteins with at least two unique peptides and H/L ratios (*Figure 6—figure supplement 3-source data 1*). Among them, nine proteins exhibited increased binding to K-Ras4a-G12V-3KR compared to K-Ras4a-G12V, and their interaction with K-Ras4a-G12V was inhibited by Sirt2 KD, suggesting that lysine defatty-acylation enhanced K-Ras4a-G12V interaction with these proteins. On the other hand, one protein showed decreased binding to K-Ras4a-G12V-3KR compared to K-Ras4a-G12V, and its

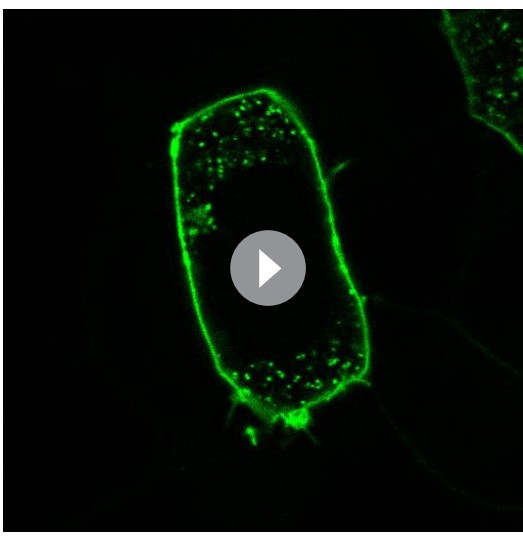

**Video 2.** Dynamics of K-Ras4a-3KR in HEK293T cells. Cells were transfected with GFP-K-Ras4a-3KR for 24 hr before being subjected to time-lapse confocal microscopy. Images were collected at 1 s intervals for 1 min of a single plane.
DOI: https://doi.org/10.7554/eLife.32436.012

interaction with K-Ras4a-G12V was increased by Sirt2 KD, suggesting that lysine defatty-acylation repressed K-Ras4a-G12V interaction with it (*Figure 6—figure supplement 3b*).

Among these 10 proteins, the serine/threonine-protein kinase A-Raf and Apoptosis-inducing factor 1 (Aif), whose interaction with K-Ras4a-G12V might be increased by lysine defatty-acylation, attracted our attention. A-Raf is a member of the Raf family of serine/threonine-specific protein kinases, acts as a Ras effector and plays an important role in apoptosis (*Rauch et al., 2010*, *2016*) and tumorigenesis (*Imielinski et al., 2014*; *Lee et al., 2010*; *Nelson et al., 2014*). In response to apoptotic stimuli, Aif is released from the mitochondrial intermembrane space into the cytosol and nucleus, where it functions as a proapoptotic factor (*Hangen et al., 2010*). Since suppression of apoptosis is linked to Ras-induced transformation (*Pylayeva-Gupta et al., 2011*), it is plausible that A-Raf and Aif are involved in the regulation of K-Ras4a transformation activity by lysine fatty acylation. To test this hypothesis, we first validated the interactome results by co-IP. Although more interaction of Aif with K-Ras4a-G12V-3KR was observed than with

K-Ras4a-G12V, Sirt2 KD did not affect the interaction of Aif with either K-Ras4a-G12V or K-Ras4a-G12V-3KR (*Figure 6—figure supplement 3c and d*) and was not investigated further. However, a greater interaction of A-Raf with K-Ras4a-G12V-3KR was observed than with K-Ras4a-G12V, and Sirt2 KD significantly decreased the interaction of A-Raf with K-Ras4a-G12V but not with K-Ras4a-G12V-3KR (*Figure 6a and b*). Thus, we concluded that A-Raf was an effector protein of K-Ras4a that

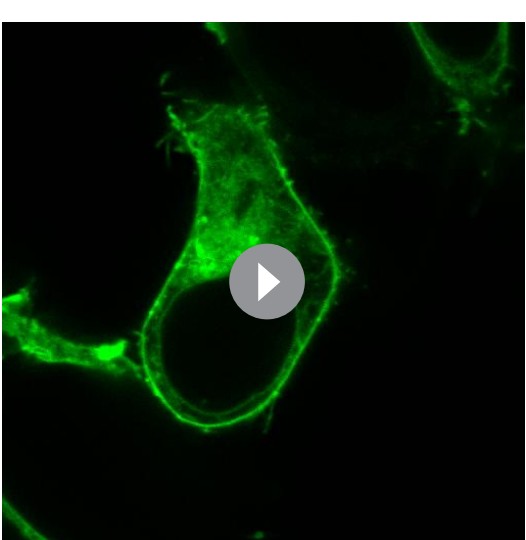

**Video 3.** Dynamics of K-Ras4a-C180S in HEK293T cells. Cells were transfected with GFP-K-Ras4a-C180S for 24 hr before being subjected to time-lapse confocal microscopy. Images were collected at 1 s intervals for 1 min of a single plane.
DOI: https://doi.org/10.7554/eLife.32436.013

was regulated by lysine fatty acylation and SIRT2. Unlike the effect of Sirt2 KD on K-Ras4a-G12V-A-Raf interaction, Sirt2 KD did not alter H-Ras-G12V-A-Raf interaction (*Figure 6a and b*). As mentioned earlier, lysine fatty acylation did not affect the binding between C-Raf (Raf1) and K-Ras4a-G12V (*Figure 6—figure supplement 2a*). We also assessed the interaction of K-Ras4a-G12V with another Raf family member, B-Raf. Co-IP indicated that neither 3KR mutation nor Sirt2 KD altered the binding of B-Raf to K-Ras4a-G12V (*Figure 6—figure supplement 2c*). These results collectively demonstrate that removal of lysine fatty acylation from K-Ras4a by SIRT2 results in its preferential association with A-Raf, but not B-Raf or C-Raf.

Our results suggest that SIRT2-mediated lysine defatty-acylation does not affect the magnitude of K-Ras4a activation but promotes endomembrane localization of active K-Ras4a. It has been reported that the efficient activation of certain effector pathways by Ras is dependent on the entry of Ras to the endosomal compartment (*Jura et al., 2006*; *Roy et al., 2002*). Therefore, it is plausible that lysine defatty-acylation may

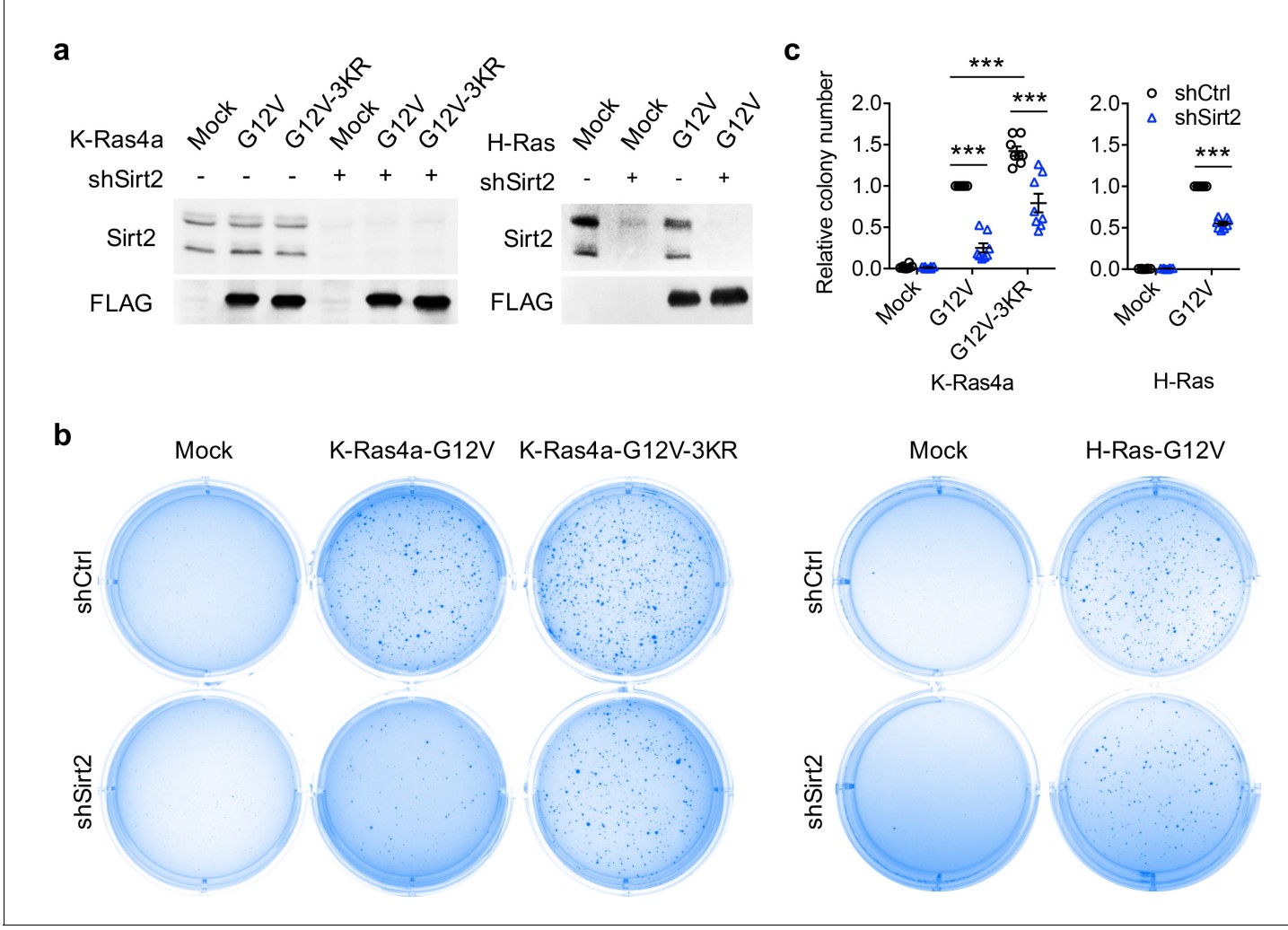

**Figure 5.** SIRT2-dependent lysine defatty-acylation increases K-Ras4a transforming activity. (a) Representative western blot analyses of Sirt2, FLAG-K-Ras4a-G12V, FLAG-K-Ras4a-G12V-3KR, FLAG-H-Ras-G12V protein levels in NIH3T3 cells with Ctrl or Sirt2 KD used in (b and c). (b) Anchorage-independent growth of NIH3T3 cells stably expressing Mock, K-Ras4a-G12V, -G12V-3KR, or H-Ras-G12V with Ctrl or Sirt2 KD. (c) Quantification of the colony numbers in (b) relative to that of the cells expressing K-Ras4a-G12V-shCtrl or H-Ras-G12V-shCtrl. Statistical evaluation was by unpaired two-tailed Student's t test. Error bars represent SEM in eight biological replicates or as indicated. ***$p < 0.001$. Representative images (a, b) from at least three independent experiments are shown.

DOI: https://doi.org/10.7554/eLife.32436.014

The following figure supplement is available for figure 5:

**Figure supplement 1.** Lysine fatty acylation regulates K-Ras4a-G12V-mediated cell proliferation but not migration.

DOI: https://doi.org/10.7554/eLife.32436.015

facilitate the endomembrane recruitment of A-Raf by K-Ras4a, thereby increasing K-Ras4a transforming activity. Live cell imaging revealed that A-Raf colocalized with K-Ras4a-G12V at both the PM and endomembranes. K-Ras4a-G12V-3KR showed more colocalization with A-Raf on the endomembranes than K-Ras4a-G12V did. Sirt2 KD inhibited the endomembrane recruitment of A-Raf by K-Ras4a-G12V but not that by K-Ras4a-G12V-3KR (*Figure 6c and d*). These results are in line with our hypothesis. Thus, it is likely that by regulating endomembrane recruitment of A-Raf, K-Ras4a lysine fatty acylation may alter its signaling output through A-Raf, thereby modulating its transforming activity.

While the functions of B-Raf and C-Raf in Ras-mediated transformation have been well elucidated, the role of A-Raf in this process remains obscure (*An et al., 2015*). So we next examined whether A-Raf plays a role in K-Ras4a-G12V mediated transformation using the soft agar colony formation

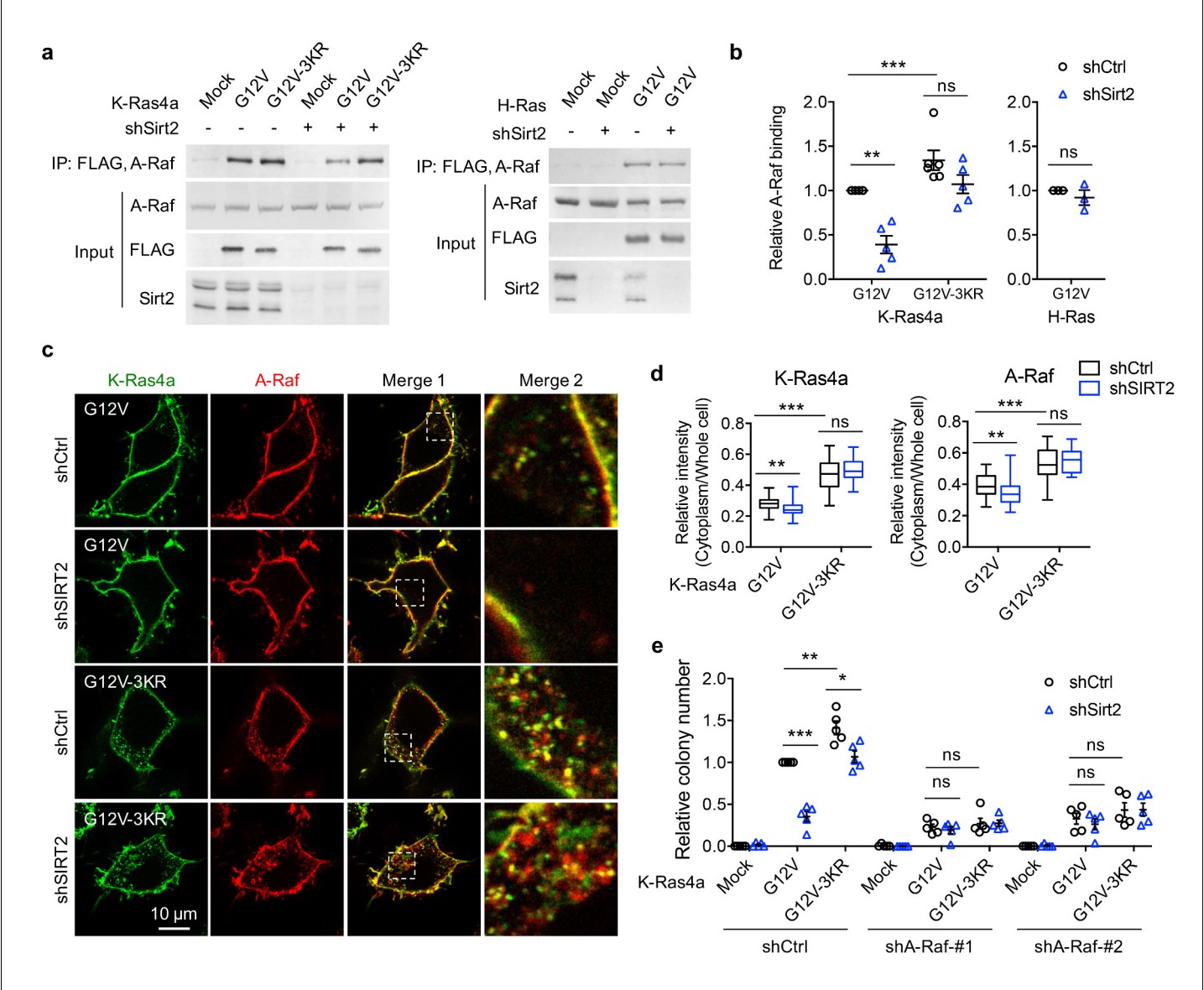

**Figure 6.** A-Raf is involved in the regulation of K-Ras4a transforming activity by lysine fatty acylation. (**a**) Co-IP of A-Raf with an anti-FLAG antibody in NIH3T3 cells stably expressing Mock, FLAG-K-Ras4a-G12V, FLAG-K-Ras4a-G12V-3KR, or FLAG-H-Ras-G12V with Ctrl or Sirt2 KD. (**b**) Quantification of relative A-Raf binding levels in (**a**). The A-Raf binding levels in Ctrl KD cells were set to 1. Quantification was done with Fiji software. Signal intensity of A-Raf was normalized with the corresponding FLAG intensity. (**c**) Images showing the localization of GFP-K-Ras4a-G12V or -G12V-3KR and DsRed-A-Raf in live HEK293T cells with Ctrl or SIRT2 KD (by shSIRT2-#2). Merge 2 shows the magnified white dashed squares-enclosed regions in Merge 1. (**d**) Statistical analyses of the relative cytoplasm to whole cell intensity of K-Ras4a and A-Raf from (**c**) (n = 17 for all samples). (**e**) Anchorage-independent growth of NIH3T3 cells stably expressing Mock, K-Ras4a-G12V or -G12V-3KR with Ctrl or Sirt2 KD, and Ctrl or A-Raf KDs. The y axis represents colony numbers relative to that of the Sirt2 and A-Raf Ctrl KD cells expressing K-Ras4a-G12V. Statistical evaluation in (**b**) and (**e**) was by unpaired two-tailed Student's t test. Error bars represent SEM in at least three biological replicates or as indicated. Statistical evaluation in (**d**) was by two-way ANOVA. *Centre line* of the box plot represents the mean value, box represents the 95% confidence interval, and whiskers represent the range of the values. *p<0.05; **p<0.01; ***p<0.001; ns, not significant. Representative images are shown.

DOI: https://doi.org/10.7554/eLife.32436.016

The following source data and figure supplements are available for figure 6:

**Figure supplement 1.** Lysine fatty acylation regulates the subcellular localization of active K-Ras4a.
DOI: https://doi.org/10.7554/eLife.32436.017

**Figure supplement 2.** Lysine fatty acylation does not affect K-Ras4a signaling through Raf1, PI3K, RalGDS, or B-Raf.
DOI: https://doi.org/10.7554/eLife.32436.018

**Figure supplement 3.** Interactome study identifies K-Ras4a-G12V interacting proteins that potentially mediate the effect of lysine fatty acylation.

*Figure 6 continued on next page*

*Figure 6 continued*

DOI: https://doi.org/10.7554/eLife.32436.019

**Figure supplement 3-source data 1.** Results from SILAC-based K-Ras4a-G12V and K-Ras4a-G12V-3KR interactome study in NIH3T3 cells.

DOI: https://doi.org/10.7554/eLife.32436.020

**Figure supplement 4.** A-Raf mediates the regulation of K-Ras4a-G12V transforming activity by lysine fatty acylation.

DOI: https://doi.org/10.7554/eLife.32436.021

assay. Inhibition of A-Raf expression with shRNA (*Figure 6—figure supplement 4a*) partially suppressed K-Ras4a-G12V-induced colony formation, indicating that A-Raf is important for K-Ras4a-G12V mediated transformation. Moreover, A-Raf KD abrogated the 3KR mutation-mediated increase and Sirt2 KD-mediated decrease in the transformation activity of K-Ras4a-G12V (*Figure 6e* and *Figure 6—figure supplement 4b*), suggesting that A-Raf is important for the regulation of K-Ras4a transforming activity by SIRT2-dependent lysine defatty-acylation. These results further support the model that lysine defatty-acylation by SIRT2 enhances the recruitment of A-Raf to K-Ras4a at endomembranes, thereby promoting transforming activity of K-Ras4a.

## Discussion

Protein lysine fatty acylation was discovered over two decades ago (*Bursten et al., 1988*; *Hedo et al., 1987*; *Pillai and Baltimore, 1987*; *Stevenson et al., 1993*; *Stevenson et al., 1992*). However, very little is known about its functional significance. Our current study furnishes a model where K-Ras4a is fatty acylated on lysine residues at its C-terminal HVR, and the removal of lysine fatty acylation by SIRT2 facilitates its endomembrane localization and interaction with A-Raf, thus enhancing its transforming activity (*Figure 7*). These findings demonstrate that K-Ras is modified and regulated by a previously under-appreciated PTM, lysine fatty acylation, which expands not only the regulatory mechanisms for Ras proteins, but also the biological significance of lysine fatty acylation. Moreover, our study reveals the first lysine defatty-acylation substrate for SIRT2 and uncovers the physiological relevance of SIRT2 as a lysine defatty-acylase (*Feldman et al., 2013*; *Liu et al., 2015*; *Teng et al., 2015*).

We found that H-Ras and K-Ras4a possess lysine fatty acylation that could be hydrolyzed by sirtuins in vitro or in cells (*Figures 1* and *2*). Although our attempt to detect N-Ras lysine fatty acylation by MS was not successful, the N-Ras-K169/170R (2KR) mutant presented decreased $NH_2OH$-resistant fatty acylation compared with WT, suggesting that N-Ras might be lysine fatty acylated (*Figure 1c* and *Figure 1—figure supplement 1b*). While cysteine palmitoylation of Ras proteins was discovered almost three decades ago (*Buss and Sefton, 1986*), lysine fatty acylation of Ras was not identified for several reasons. First, lysine fatty acylation did not emerge as a physiologically significant modification until recent years. Correspondingly, the possibility of lysine fatty acylation on Ras proteins had not been investigated previously. Second, previously people only focused on Ras cysteine palmitoylation because mutations of the palmitoylated cysteine to serine abolished the palmitoylation of H-Ras, N-Ras (*Hancock et al., 1989*), and K-Ras4a (*Tsai et al., 2015*) based on [3]H-palmitic acid labeling. Therefore, lysine fatty acylation of Ras proteins might have been missed based on the mutagenesis results. Similar to, but slightly different from these previous reports, we found that mutating the palmitoylated cysteine of K-Ras4a decreased lysine fatty acylation by nearly 90% (*Figure 3c*) but not completely. A similar effect of the palmitoylated cysteine to serine mutation was also observed for R-Ras2 (*Zhang et al., 2017*). Last, although the palmitoylation sites for Ras proteins were characterized with chemoproteomic approaches based on acyl-biotin exchange (ABE) (*Drisdel and Green, 2004*; *Kang et al., 2008*) or acyl-resin-assisted capture (acyl-RAC) (*Forrester et al., 2011*; *Lanyon-Hogg et al., 2017*), these approaches are cysteine-centric and do not allow identification of amide-linked fatty acylation. Direct site identification of palmitoylation has been challenging owing to the low abundance and high hydrophobicity of modified peptides, which are easily lost during sample preparation (*Peng et al., 2016*). Our current study highlights the regulation of Ras proteins by lysine fatty acylation and suggests that additional studies are required to understand the regulation of this important class of proteins. Many members of the Ras superfamily of GTPases contain lysine-rich sequences at their C-termini. It is therefore of great interest to us to

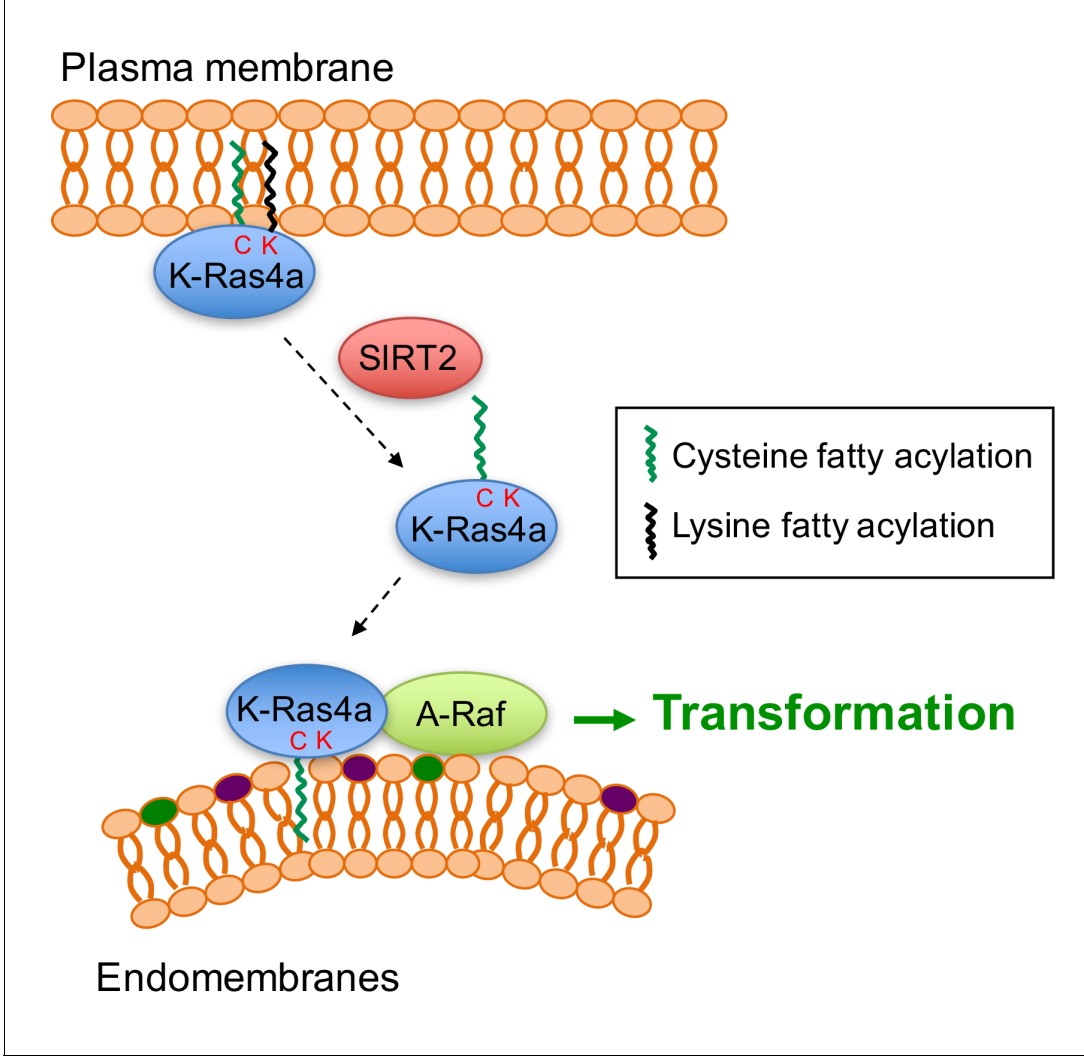

**Figure 7.** Model for the regulation of K-Ras4a by SIRT2-mediated removal of lysine fatty acylation. Removal of lysine fatty acylation by SIRT2 facilitates K-Ras4a to localize to endomembranes and interact with A-Raf, and thus enhances its activity to transform cells.
DOI: https://doi.org/10.7554/eLife.32436.022

investigate whether lysine fatty acylation could act as a general regulatory mechanism for many Ras-related small GTPases.

The discovery of lysine fatty acylation on K-Ras4a raises the question of the relative abundance of lysine versus cysteine fatty acylation. Semi-quantification of the fluorescence intensity from Alk14 labeling results enables us to roughly estimate the stoichiometry of lysine fatty acylation. Based on this, K-Ras4a exhibits nearly 50% of lysine fatty acylation relative to total fatty acylation (*Figure 1c*). Therefore, the ratio of cysteine palmitoylation to lysine fatty acylation may be close to 1:1 on K-Ras4a. The 3KR mutation decreased K-Ras4a lysine fatty acylation by about 50% (*Figure 1f and 3e*), suggesting that the C-terminal lysine fatty acylation regulated by SIRT2 is around 50% of the lysine fatty acylation and 25% of the total fatty acylation. Regarding endogenous K-Ras4a, by quantifying the K-Ras4a western blot signal from the streptavidin beads and supernatant in *Figure 2i*, we estimated that about 28% and 50% of the fatty acylated K-Ras4a is lysine fatty acylated in Ctrl KD and SIRT2 KD HCT116 cells, respectively. Unfortunately, precise quantitation of protein fatty acylation still remains a significant unsolved challenge and we could not determine the ratio of fatty acylated versus unmodified K-Ras4a.

To study the physiological function of K-Ras4a lysine fatty acylation, we utilized the K-Ras4a-3KR mutant in combination with SIRT2 KD. The lysine-to-arginine mutant maintains the positive charge of

the polybasic patch, which makes it a good lysine fatty acylation-deficient mimic. Recently, Zhou *et al.* reported that lysine and arginine residues are not equivalent in determining the membrane lipid binding specificity of K-Ras4b C-terminus, which raises the possibility that the effect of 3KR mutation might not be solely due to lack of lysine fatty acylation (*Zhou et al., 2017*). Likewise, changes in the SIRT2 KD cells could be mediated through other substrates for SIRT2. Therefore, it is critical to employ both the 3KR mutant and SIRT2 KD to rule out these possibilities. SIRT2 KD enhances the lysine fatty acylation of K-Ras4a-WT but not the 3KR mutant. Thus, if a biological effect is due to lysine fatty acylation, SIRT2 should have a greater impact on the effect of K-Ras4a-WT than that of the 3KR mutant. Of note, H-Ras, which shares similar properties with K-Ras4a, but is not a lysine defatty-acylase target for SIRT2, also serves as a good control for the effect of SIRT2 KD. Indeed, SIRT2 KD repressed the endomembrane localization, transforming activity, and A-Raf binding of K-Ras4a-WT more than that of K-Ras4a-3KR or H-Ras, indicating that SIRT2-dependent lysine defatty-acylation facilitates endomembrane localization of K-Ras4a, enhances its interaction with A-Raf, and thus promotes cellular transformation.

Goodwin *et al.* previously reported that cysteine depalmitoylated H-Ras and N-Ras traffic to and from the Golgi complex by a nonvesicular mechanism, and suggested a model where cysteine palmitoylation traps Ras on membranes, enabling Ras to undergo vesicular transport (*Goodwin et al., 2005*). In line with this, Tsai *et al.* (*Tsai et al., 2015*) and we observed that the K-Ras4a-C180S mutant, which possesses no cysteine palmitoylation and little lysine fatty acylation, localizes to ER/Golgi-like internal membranes (*Figure 4—figure supplement 1f and g*). Differently, we found that removal of the lysine fatty acylation by SIRT2, which results in K-Ras4a with only cysteine palmitoylation, promotes endomembrane localization of K-Ras4a (*Figure 4a*). This evidence supports the model that cysteine palmitoylation enables K-Ras4a to undergo vesicular transport, whereas lysine fatty acylation blocks K-Ras4a translocation from the plasma membrane to endomembranes (*Figure 7*). Furthermore, the C180S mutant suppressed K-Ras4a-G12V-mediated anchorage-independent growth and activation of MAPK signaling (*Tsai et al., 2015*; *Zhao et al., 2015*). In contrast, the 3KR mutant increased K-Ras4a-G12V-mediated anchorage-independent growth (*Figure 5b and c*), exhibited no effect on MAPK signaling (*Figure 6—figure supplement 2*), but activated A-Raf instead (*Figure 6a*). Considering the possibility that lysine fatty acylation largely relies on cysteine palmitoylation to occur, it is likely that the reversible lysine fatty acylation adds a layer of regulation for K-Ras4a above that of cysteine palmitoylation.

We noted that K-Ras4a-G12V (*Figure 6c*) exhibits weaker endomembrane localization than K-Ras4a (*Figure 4a*), which raises questions about the differential regulation of K-Ras4a and K-Ras4a-G12V subcellular localization. K-Ras4a and K-Ras4a-G12V possess comparable total and lysine fatty acylation levels (*Figure 4—figure supplement 1c*), suggesting that the difference in their subcellular distribution is not due to fatty acylation. It is possible that the activation state of K-Ras4a influences its dynamics on the plasma membrane and endomembranes, the mechanism for which remains to be elucidated in the future.

In the GTP-bound active form, Ras proteins bind directly to the Ras binding domain (RBD) of Raf, then form secondary interactions with a cysteine-rich domain (CRD). Although the RBD and CRD are highly conserved in all Raf isozymes, there is evidence for different binding affinities for Ras proteins to the individual Raf proteins (*Matallanas et al., 2011*; *Wellbrock et al., 2004*). For example, Weber *et al.* reported that A-Raf presents significantly lower binding affinities to H-Ras-G12V as compared to C-Raf because the Ras-binding interface of C-Raf differs from A-Raf by a conservative arginine to lysine exchange at residue 59 and 22, respectively (*Weber et al., 2000*). Furthermore, Williams *et al.* (*Williams et al., 2000*) and Fischer *et al.* (*Fischer et al., 2007*) found that farnesylation of H-Ras is required for its binding to C-Raf but not to B-Raf, implying the involvement of Ras C-terminal PTM in regulating Ras-Raf interactions. Based on these previous studies, it is likely that the C-terminal PTM of K-Ras4a regulates its interaction with Raf isozymes and that lysine fatty acylation may inhibit the binding of K-Ras4a to A-Raf but not to B-Raf and C-Raf.

Mutations that activate Ras are found in about 30% of all human tumors screened. *KRAS* mutations, which affects both K-Ras4a and K-Ras4b, occur most frequently, accounting for 86% of *RAS*-driven cancers (*Cox et al., 2014*). Although K-Ras4a is homologous to the transforming cDNA identified in Kirsten rat sarcoma virus (*Shimizu et al., 1983*), its function and regulation is less characterized compared to K-Ras4b. Recent studies showed that K-Ras4a is widely expressed in human cancers, suggesting that K-Ras4a plays a significant role in *KRAS*-driven tumors (*Tsai et al., 2015*;

*Zhao et al., 2015*). Our findings reveal that K-Ras4a is regulated by SIRT2-dependent lysine defatty-acylation. Depletion of SIRT2 increased lysine fatty acylation and diminished transforming activity of K-Ras4a, suggesting that interference with K-Ras4a lysine fatty acylation could be an approach to anti-K-Ras therapy.

The seven mammalian sirtuins, SIRT1-7, are implicated in various biological pathways and are considered potential targets against a number of human diseases (*Jing and Lin, 2015*). So far, the known biological functions of sirtuins have been mainly attributed to their deacetylase activities. Although sirtuins are increasingly recognized as lysine deacylases in addition to deacetylases, the biological significance of sirtuins as lysine deacylases remains largely unknown (*Bheda et al., 2016*). Our work here identifies the first physiological defatty-acylation substrate for SIRT2. Since protein acyl lysine modifications likely use acyl-CoA molecules as the acyl donors, the cellular metabolic state can affect acyl lysine PTMs by altering the concentration of acyl-CoA molecules. Sirtuins requires NAD as a co-substrate and the NAD level is regulated by cellular metabolism. Thus SIRT2 may provide an additional link between K-Ras4a signaling and cellular metabolism. Given that Ras proteins play critical roles in many human cancers, SIRT2, as a Ras regulator, may be an important therapeutic target for cancer, which is consistent with several recent reports (*Jing et al., 2016*; *Moniot et al., 2017*; *Shah et al., 2016*; *Wang et al., 2014*; *Wilking-Busch et al., 2017*; *Xu et al., 2016*; *Zhao et al., 2013*, *2016*). The physiological and pathophysiological roles of SIRT2 thus merit further investigation.

# Materials and methods

**Key resources table**

| Reagent type (species) or resource | Designation | Source or reference | Identifiers | Additional information |
|---|---|---|---|---|
| cell line (human) | HEK293T | ATCC | Cat#: CRL-3216, RRID: CVCL_0063 | |
| cell line (human) | HCT116 | ATCC | Cat#: CCL-247, RRID: CVCL_0291 | |
| cell line (mouse) | NIH3T3 | ATCC | Cat#: CRL1658, RRID: CVCL_0594 | |
| cell line (human) | MEF *Sirt2* WT and KO | Other | N/A | Obtained from Dr. Toren Finkel's laboratory at NIH, Bethesda, MD |
| transfected construct | pLKO.1-shLuciferase | Sigma-Aldrich | SHC007 | |
| transfected construct (human) | pLKO.1-shSIRT1-#1 | Sigma-Aldrich | TRCN0000018980 | |
| transfected construct (human) | pLKO.1-shSIRT1-#2 | Sigma-Aldrich | TRCN0000018981 | |
| transfected construct (human) | pLKO.1-shSIRT2-#1 | Sigma-Aldrich | TRCN0000040219 | |
| transfected construct (human) | pLKO.1-shSIRT2-#2 | Sigma-Aldrich | TRCN0000310335 | |
| transfected construct (mouse) | pLKO.1-shSIRT2 | Sigma-Aldrich | TRCN0000012118 | |
| transfected construct (mouse) | pLKO.1-shA-Raf-#1 | Sigma-Aldrich | TRCN0000022612 | |
| transfected construct (mouse) | pLKO.1-shA-Raf-#2 | Sigma-Aldrich | TRCN0000022610 | |
| transfected construct (human) | pCMV5-FLAG-K-Ras4a | This paper | N/A | Cloning described in *Cloning and mutagenesis* |
| transfected construct (human) | pCMV5-FLAG-H-Ras | This paper | N/A | Provided by Dr. Maurine E. Linder |
| transfected construct (human) | pCMV5-FLAG-N-Ras | This paper | N/A | Provided by Dr. Maurine E. Linder |

*Continued on next page*

*Continued*

| Reagent type (species) or resource | Designation | Source or reference | Identifiers | Additional information |
|---|---|---|---|---|
| transfected construct (human) | pCMV5-FLAG-K-Ras4b | This paper | N/A | Provided by Dr. Maurine E. Linder |
| transfected construct (human) | pCMV5-FLAG-RalA | (*Nishimura and Linder, 2013*) | N/A | Provided by Dr. Maurine E. Linder |
| transfected construct (human) | pAcGFP-K-Ras4a | This paper | N/A | Cloning described in *Cloning and mutagenesis.* |
| transfected construct (human) | pAcGFP-H-Ras | This paper | N/A | Cloning described in *Cloning and mutagenesis* |
| transfected construct (human) | pCDH-FLAG-K-Ras4a | This paper | N/A | Cloning described in *Cloning and mutagenesis* |
| transfected construct (human) | pCDH-FLAG-H-Ras | This paper | N/A | Cloning described in *Cloning and mutagenesis* |
| transfected construct (human) | pCMV-tag-4a-SIRT2-FLAG | This paper | N/A | Cloning described in *Cloning and mutagenesis* |
| transfected construct (human) | pCMV-tag-4a-DsRed-A-Raf | This paper | N/A | Cloning described in *Cloning and mutagenesis* |
| transfected construct (human) | DsRed-GalT | (*Luo et al., 2015*) | N/A | Obtained from Dr. Yuxin Mao at Cornell University, Ithaca, NY |
| transfected construct (human) | mCherry-Sec61 beta | Addgene (*Zurek et al., 2011*) | #49155 | |
| transfected construct (human) | mCherry-Rab11 | Addgene (*Choudhury et al., 2002*) | #55124 | |
| transfected construct (human) | DsRed-Rab7 | Addgene (*Choudhury et al., 2002*) | #12661 | |
| transfected construct (human) | Lamp1-RFP | Addgene (*Sherer et al., 2003*) | #1817 | |
| antibody | anti-Ras (Y13-259) | EMD Chemicals Inc. | Cat#: OP01A, RRID: AB_10681741 | 1:5000 dilution for WB, 1 μg/1 mg total protein for IP |
| antibody | anti-RalA | EMD Chemicals Inc. | Cat#: ABS223, RRID: AB_11204894 | 1:1000 dilution for WB |
| antibody | anti-SIRT1 | EMD Chemicals Inc. | Cat#: 05–1243, RRID: AB_1163501 | 1:1000 dilution for WB |
| antibody | anti-SIRT2 | Abcam | ab134171 | 1:1000 dilution for WB |
| antibody | anti-SIRT2 | Cell Signaling Technology | 12650 | 1:1000 dilution for WB |
| antibody | anti-Transferrin Receptor | Abcam | Cat#: ab84036, RRID: AB_10673794 | 1:2000 dilution for WB |
| antibody | anti-K-Ras4a | Santa Cruz Biotechnology | Cat#: sc-522 RRID: AB_2134128 | 1:250 dilution for WB |
| antibody | anti-A-Raf | Santa Cruz Biotechnology | Cat#: sc-408 RRID: AB_630882 | 1:1000 dilution for WB |
| antibody | anti-B-Raf | Santa Cruz Biotechnology | Cat#: sc-166 RRID: AB_630938 | 1:5000 dilution for WB |
| antibody | anti-Raf1 | Santa Cruz Biotechnology | Cat#: sc-227 RRID: AB_632303 | 1:1000 dilution for WB |
| antibody | anti-FLAG M2 conjugated with HRP | Sigma-Aldrich | Cat#: A8592 RRID: AB_439702 | 1:5000 dilution for WB |
| antibody | anti-FLAG M2 affinity gel | Sigma-Aldrich | Cat#: A2220 RRID: AB_10063035 | 10 μL slurry/1 mg total protein for IP |
| antibody | anti-Syntaxin 6 | Cell Signaling Technology | Cat#: 2869 RRID: AB_2196500 | 1:50 dilution for IF |
| antibody | EEA1 | Cell Signaling Technology | Cat#: 3288 RRID: AB_2096811 | 1:100 dilution for IF |

*Continued on next page*

*Continued*

| Reagent type (species) or resource | Designation | Source or reference | Identifiers | Additional information |
|---|---|---|---|---|
| recombinant protein | SIRT1 | (*Du et al., 2009*) | N/A | |
| recombinant protein | SIRT2 | (*Jing et al., 2016*) | N/A | |
| recombinant protein | SIRT3 | (*Du et al., 2009*) | N/A | |
| recombinant protein | SIRT6 | (*Jiang et al., 2013*) | N/A | |
| peptide | acetyl H3K9 | (*Zhu et al., 2012*) | N/A | |
| peptide | myristoyl H3K9 | (*Zhu et al., 2012*) | N/A | |
| peptide | myristoyl K-Ras4a-C180 | This paper | N/A | Described in *HPLC-based SIRT2 activity assay* |
| peptide | triple FLAG peptide | Sigma-Aldrich | F4799 | |
| commercial assay or kit | Active Ras Pull-Down and Detection Kit | Thermo Fisher Scientific | 16117 | |
| chemical compound, drug | Alk14 | (*Charron et al., 2009*) | N/A | |
| chemical compound, drug | Rhodamine-azide | (*Charron et al., 2009*) | N/A | |
| chemical compound, drug | Azide-PEG3-Biotin | Sigma-Aldrich | 762024 | |
| chemical compound, drug | Tris[(1-benzyl-1H-1,2,3-triazol-4-yl)methyl]amine (TBTA) | Sigma-Aldrich | 678937 | |
| chemical compound, drug | Tris(2-carboxyethyl) phosphine (TCEP) | Sigma-Aldrich | 75259 | |
| chemical compound, drug | Hydroxylamine | Sigma-Aldrich | 159417 | |
| chemical compound, drug | L-lysine | Sigma-Aldrich | L9037 | |
| chemical compound, drug | L-arginine | Sigma-Aldrich | A8094 | |
| chemical compound, drug | $[^{13}C_6, ^{15}N_2]$-L-lysine | Sigma-Aldrich | 608041 | |
| chemical compound, drug | $[^{13}C_6, ^{15}N_4]$-L-arginine | Sigma-Aldrich | 608033 | |
| chemical compound, drug | $^{32}$P-NAD$^+$ | PerkinElmer | BLU023 $\times$ 250UC | |
| Software | Fiji | N/A | https://fiji.sc/ RRID: SCR_002285 | |
| other | FuGene 6 | Promega | E2692 | |
| other | Sequencing grade modified trypsin | Promega | V5111 | |

## Common reagents and antibodies

Chemicals from commercial sources were obtained in the highest purity available. Trichostatin A (TSA, T8552), protease inhibitor cocktail (P8340), phosphatase inhibitor cocktail (P0044), NAD (NAD100-RO), Puromycin (P8833), Crystal Violet (C0775), low-melting point agarose (A0701) were purchased from Sigma-Aldrich (St. Louis, MO). The anti-Phospho-Erk1/2 (Thr202/204) (9101), Erk1/2 (4696), Phospho-Akt (Thr308) (9275), Phospho-Akt (Ser473) (9271), Akt (4691), Phospho-SAPK/JNK (Thr182/Tyr185) (4668), SAPK/JNK (9252) antibodies were purchased from Cell Signaling Technology (Danvers, MA). The anti-β-Actin (sc-4777), Na/K-ATPase (sc-21712), GAPDH (sc-47724), the normal rat IgG (sc-2026) and the goat anti-mouse/rabbit/Rat IgG-horseradish peroxidase-conjugated antibodies were purchased from Santa Cruz Biotechnology (Dallas, TX). The anti-Acetyl Lysine antibody (ICP0380) was obtained from ImmuneChem (Cananda). Enzyme-linked chemiluminescence (ECL) plus (32132) western blotting detection reagent, Cy3-conjugated goat anti-rabbit IgG (H + L) secondary antibody (A10520) and the high capacity Streptavidin agarose (20357) were purchased from Thermo Fisher Scientific (Waltham, MA). Saponin (S0019-25G) was from TCI America (Portland, OR). Sep-Pak C18 cartridge was purchased from Waters (Milford, MA).

## Cloning and mutagenesis

The human K-RAS4A expression vector with N-terminal FLAG tag was obtained by RT-PCR amplification of *K-RAS4A* and subcloning *via* EcoRI and SalI sites into pCMV5 vector. The human *K-RAS4A* lentiviral vector was obtained by inserting FLAG-*K-RAS4A* into pCDH-CMV-MCS-EF1-Puro vector between the EcoRI and NotI sites. The human *H-RAS* lentiviral vector was obtained by inserting FLAG-*H-RAS* into pCDH-CMV-MCS-EF1-Puro vector between the EcoRI and BamHI sites. The GFP-K-Ras4a and GFP-H-Ras expression vectors were constructed by inserting *K-RAS4A and H-RAS* cDNA into pAcGFP-C1 vector between the BglII and SalI sites, respectively. The human *STX6* expression vector with N-terminal FLAG tag was constructed by inserting FLAG-STX6 cDNA into pCMV-tag-4a vector between the EcoRI and XhoI sites. To generate the expression vector for human SIRT2 with C-terminal FLAG tag, full-length human *SIRT2* cDNA was amplified by PCR and inserted into pCMV-tag-4a vector between the BamHI and XhoI sites. The expression vectors for H-RAS, N-RAS, K-RAS4A mutants and SIRT2-H187Y were generated by QuikChange site-directed mutagenesis (*Liu and Naismith, 2008*). The DsRed cDNA without stop codon was inserted using NotI and BamHI sites into pCMV-tag-4a to generate pCMV-tag-4a-DsRed-C vector that enables cloning of gene of interest with N-terminal DsRed. The DsRed-RBD expression vector was constructed by inserting cDNA coding the Ras-binding domain of human Raf1 (aa51-131) into pCMV-tag-4a-DsRed-C vector between EcoRV and XhoI sites. The DsRed-A-Raf expression vector was generated by inserting mouse *Araf* cDNA into pCMV-tag-4a-DsRed-C vector using BamHI and EcoRI sites.

## Cell culture, transfection and transduction

Human HEK293T cells were grown in DMEM media (11965–092, Gibco) with 10% heat inactivated (HI) fetal bovine serum (FBS, 26140079, Gibco). Mouse embryonic fibroblast NIH3T3 cells and *Sirt2* WT and KO MEF cells were grown in DMEM media supplemented with non-essential amino acids (11140050, Gibco) and 15% HI FBS. Human HCT116 cells were grown in McCoy's 5A media (16600082, Gibco) with 10% HI FBS. Sirt2 WT and KO MEF cells were obtained from Dr. Toren Finkel's laboratory at NIH and the knockout of Sirt2 was confirmed by Western blot. All other cell lines were purchased from ATCC and were not further authenticated after purchase from ATCC. All cell lines were tested for and showed no mycoplasma contamination.

For SILAC experiments, 'light' NIH3T3 cells were maintained in DMEM media for SILAC (88420, Thermo Fisher Scientific) supplemented with 100 mg/L [$^{12}C_6$, $^{14}N_2$]-L-lysine, 100 mg/L [$^{12}C_6$, $^{14}N_4$]-L-arginine, non-essential amino acids, and 15% dialyzed FBS (26400036, Thermo Fisher Scientific); 'heavy' NIH3T3 cells were cultured in DMEM media for SILAC supplemented with 100 mg/L [$^{13}C_6$, $^{15}N_2$]-L-lysine, 100 mg/L [$^{13}C_6$, $^{15}N_4$]-L-arginine, non-essential amino acids, and 15% dialyzed FBS. Cells were cultured in SILAC media for at least six doubling times to achieve maximum incorporation of 'labeled' amino acids into proteins before the interactome study was performed.

To transiently overexpress proteins of interest in cells, the expression vectors were transfected into cells using FuGene 6 according to the manufacturer's protocol. Empty vector was transfected as a negative control. Lentiviral infection for overexpressing H-Ras, K-Ras4a-WT and mutants or knocking down SIRT1, SIRT2 and A-Raf was performed as previously described (*Jiang et al., 2013*; *Jing et al., 2016*). Puromycin (3 μg/mL for NIH3T3 cells, 1.5 μg/mL for HEK293T cells) was added to the cell culture media to select NIH3T3 cells with stable overexpression of Mock (pCDH empty vector control), K-Ras4a-G12V, K-Ras4a-G12V-3KR or H-Ras-G12V as well as HEK293T cells with stable luciferase KD (Ctrl KD), SIRT1 KD, or SIRT2 KD.

## Immunoprecipitation of Alk14-labeled proteins of interest

HEK293T cells (parental cells, luciferase KD, SIRT1 KD, or SIRT2 KD) were transiently transfected to express FLAG-tagged protein of interest overnight. The cells were then cultured with fresh medium containing 50 μM Alk14 for 6 hr. Cells were collected and lysed in 1% NP-40 lysis buffer (25 mM Tris-HCl pH 8.0, 150 mM NaCl, 10% glycerol, 1% Nonidet P-40) with protease inhibitor cocktail. The supernatant was collected after centrifugation at 16,000 g for 20 min at 4°C. Protein concentration was determined by Bradford assay (23200, Thermo Fisher Scientific). 0.5–1 mg cell lysate was incubated with 10 μL suspension of anti-FLAG M2 affinity gel for 2 hr at 4°C. The affinity gel was then centrifuged at 500 *g* for 2 min at 4°C, washed three times with 1 mL IP washing buffer (25 mM Tris-HCl pH 8.0, 150 mM NaCl, 0.2% Nonidet P-40) and used for further experiments.

## Detection of fatty acylation on protein of interest by on-beads click chemistry and in-gel fluorescence

The immunopurified protein with Alk14 labeling was suspended in 14 µL IP washing buffer for click chemistry. Rh-N$_3$ (3 µL of 1 mM solution in DMF, final concentration 150 µM) was added to the above suspension, followed by the addition of TBTA (1 µL 10 mM solution in DMF, final concentration 500 µM), CuSO$_4$ (1 µL of 40 mM solution in H$_2$O, final concentration 2 mM), and TCEP (1 µL of 40 mM solution in H$_2$O, final concentration 2 mM). The click chemistry reaction was allowed to proceed at room temperature for 30 min. The reaction mixture was mixed with 10 µL of 6 × protein loading buffer and heated at 95°C for 10 min. After centrifugation at 16,000 g for 2 min at room temperature, 15 µL of the supernatant was treated with NH$_2$OH (pH 8.0, 1 µL of 5 M solution in H$_2$O, final concentration 312 mM) or equivalent volume of water (negative control) at 95°C for 7 min. The samples were resolved by SDS-PAGE. Rhodamine fluorescence signal was recorded by Typhoon 9400 Variable Mode Imager (GE Healthcare Life Sciences, Piscataway, NJ) with setting of Green (532 nm)/580BP30 PMT 500 V (normal sensitivity). Fiji software (*Schindelin et al., 2012*) was used for quantification of the fluorescence intensity. Signal intensity of in-gel fluorescence was normalized with respect to that of the corresponding FLAG western blot.

## Defatty-acylation of K-Ras4a by sirtuins in vitro

The *Plasmodium falciparum* Sir2A (PfSir2A) (*Zhu et al., 2012*), the human SIRT1 (*Du et al., 2009*), SIRT2 (*Jing et al., 2016*), SIRT3 (*Du et al., 2009*) and SIRT6 (*Jiang et al., 2013*) were expressed as previously described. The immunoprecipated Ras with Alk14 labeling on anti-FLAG affinity gel was suspended in 25 µl of assay buffer (50 mM Tris-HCl, pH 8.0, 100 mM NaCl, 2 mM MgCl$_2$, 1 mM DTT) with 10 µM of PfSir2A or 5 µM of SIRT1, SIRT2, SIRT3, SIRT6 or the corresponding amount of BSA and with or without 1 mM NAD and incubated at 37°C for 30 min (SIRT2) or 1 hr (PfSir2A, SIRT1, 3 and 6). The reaction was stopped by washing the affinity gel using 1 mL of IP washing buffer for three times. On-bead click chemistry and in-gel fluorescence were carried out as described above.

## High-performance liquid chromatography (HPLC)-based SIRT2 activity assay

SIRT2 or SIRT2-H187Y (1 µM) was incubated in 60 µL of reaction buffer (20 mM Tris, pH 8.0, 1 mM DTT, 1 mM NAD) with 32 µM acetyl H3K9, myristoyl H3K9, or myristoyl K-Ras4a-C180 peptides, respectively, at 37°C for 10 min (deacetylation) or 20 min (demyristoylation). Reactions were quenched with 60 µL ice-cold acetonitrile and spun down at 18,000 g for 10 min to remove the precipitated protein. The supernatant was then analyzed by HPLC on a Kinetex XB-C18 column (100 A, 75 mm ×4.6 mm, 2.6 µm, Phenomenex). The peak areas were integrated and the conversion rate was calculated from the ratio of the free H3K9 peptide peak area over the total peak areas of the substrate and product peptides.

## Western blot

Western blot analysis was performed as described previously (*Jiang et al., 2013*). The proteins of interest were detected using ECL plus and visualized using the Typhoon 9400 Variable Mode Imager (GE Healthcare). Quantification of signal intensity from western blots was done using Fiji software. To assess the effect of lysine fatty acylation on the signaling output of K-Ras4a-G12V through Erk, Akt and Jnk, NIH3T3 cells stably expressing Mock, FLAG-K-Ras4a-G12V or -G12V-3KR were infected with lentivirus carrying luciferase (Ctrl) or mouse Sirt2 shRNA for 3 days, collected and lysed in 1% NP-40 lysis buffer with protease inhibitor cocktail and phosphatase inhibitor cocktail. Cell lysates were then subjected to western blot for the analyses of indicated proteins.

To detect acetyl lysine on K-Ras4a, HEK293T cells with stable Ctrl KD or SIRT2 KD were transfected with empty vector or pCMV5-FLAG-K-Ras4a overnight. The cells were then treated with ethanol or trichostatin A (TSA, 1 µM) for 1 hr. The cells were collected and lysed in 1% NP-40 lysis buffer with protease inhibitor cocktail. Cell lysates (~3 mg), with/without overexpression of K-Ras4a, were incubated with 10 µL of anti-FLAG M2 affinity gel suspension for 2 hr at 4°C. The affinity gel was washed three times with 1 mL of IP washing buffer and then heated in 15 µL of 2 × protein loading buffer at 95°C for 10 min. The supernatant was then resolved by SDS-PAGE and the acetylation of K-Ras4a was examined by western blot using anti-acetyllysine antibody after transfer to a PVDF

membrane. Total cell lysates from TSA-treated HEK293T cells were used as a positive control for the acetyllysine blot. After recording the acetyl-lysine signal, the PVDF membrane was stained with Coomassie blue to detect K-Ras4a protein. A western blot using anti-FLAG antibody was carried out in parallel to demonstrate equal loading of K-Ras4a.

## Subcellular fractionation

HEK293T cells were transfected with pCMV5-FLAG-K-Ras4a and cultured for overnight before being collected. Cell pellets were re-suspended in subcellular fraction buffer (250 mM Sucrose, 20 mM HEPES, pH 7.4, 10 mM KCl, 1.5 mM $MgCl_2$, 1 mM EDTA, 1 mM EGTA and 1 mM DTT) containing protease inhibitor cocktail and homogenized on ice by 10 passes through a 25-gauge syringe needle. Nuclei and intact cells were removed by centrifugation at 3,000 rpm for 5 min. The mitochondrial fraction was removed by centrifuging the postnuclear supernatant at 8,000 rpm for 5 min. The supernatant was ultracentrifuged at 40,000 rpm for 1 hr. The resulting supernatant (cytosol fraction) was concentrated by ultrafiltration. The pellet (membrane fraction) was washed with subcellular fraction buffer, re-centrifuged for 45 min and dissolved in 4% SDS lysis buffer (4% SDS, 50 mM triethanolamine pH 7.4, and 150 mM NaCl). Equivalent portions of the cytosol and membrane fractions were then subjected to western blot analyses.

## Co-Immunoprecipitation

To examine the interaction between FLAG-tagged K-Ras4a and SIRT2, HEK293T cells transfected with empty vector or pCMV5-FLAG-K-Ras4a were cultured overnight, collected and lysed in 1% NP-40 lysis buffer with protease inhibitor cocktail. To examine the interaction between FLAG-tagged K-Ras4a-G12V/K-Ras4a-G12V-3KR or H-Ras-G12V and A-Raf/B-Raf/C-Raf (Raf1)/p110α/RalGDS, NIH3T3 cells stably expressing Mock, FLAG-K-Ras4a-G12V, -G12V-3KR, or FLAG-H-Ras-G12V were infected with lentivirus carrying luciferase (Ctrl) or mouse Sirt2 shRNA for 3 days, collected and lysed in 1% NP-40 lysis buffer with protease inhibitor cocktail. For both experiments, total cell lysates (2 mg of total protein for detecting SIRT1/2, 50 µg for A-Raf and C-Raf, 1 mg for B-Raf, p110α and RalGDS, determined by Bradford assay) were incubated with 10 µL suspension of anti-FLAG M2 affinity gel for 2 hr at 4°C. The resulting affinity gel was washed three times with 1 mL IP washing buffer and heated in protein loading buffer (2 × final concentration) at 95°C for 10 min. Western blot was then performed to detect levels of the indicated proteins.

## Detection of lysine fatty acylation on K-Ras4a using the [32]P-NAD assay

The [32]P-NAD assays were carried out as described previously with minor modification (*Du et al., 2011*). HEK293T cells were transfected with empty pCMV5 vector or pCMV5-K-Ras4a overnight and lysed in 1% NP-40 lysis buffer with protease inhibitor cocktail. For each reaction, cell lysates (3 mg of total protein, determined by Bradford assay) were incubated with 10 µL suspension of anti-FLAG M2 affinity gel for 2 hr at 4°C. The affinity gel was washed three times with 1 mL of IP washing buffer. The resulting anti-FLAG affinity gel or the synthetic acetyl and myristoyl H3K9 peptides (*Du et al., 2011*) (25 µM, positive control) were mixed with 10 µL solutions containing 1 µCi [32]P-NAD, 50 mM Tris-HCl pH 8.0, 150 mM NaCl, 1 mM DTT. The reactions were incubated with 1 µM BSA (negative control), SIRT2, or SIRT2-H187Y at 37°C for 30 min. A total of 2 µL of each reaction were spotted onto silica gel TLC plates and developed with 7:3 ethanol:ammonium bicarbonate (1 M aqueous solution). After development, the plates were air-dried and exposed to a PhosphorImaging screen (GE Healthcare). The signal was detected using Typhoon 9400 Variable Mode Imager (GE Healthcare).

## Biotin pull-down of lysine fatty acylated endogenous K-Ras4a

The assay was carried out as previously described with some modifications (*Wilson et al., 2011*). Briefly, HCT116 cells were infected with lentivirus carrying luciferase (Ctrl) or SIRT2 shRNA for 3 days and treated without or with Alk14 (50 µM) for 6 hr before being collected. Total proteins were then extracted using 1% NP-40 lysis buffer with protease inhibitor cocktail. 10 mg of total protein extract was subjected to click reaction with 100 µM Biotin-$N_3$, 500 µM TBTA, 1 mM $CuSO_4$ and 1 mM TCEP in a final volume of 5 mL. The reaction was allowed to proceed at room temperature for 1 hr. Proteins were precipitated by adding 4 volumes of ice-cold methanol, 3 volumes of water, and 1.5

volumes of chloroform. Precipitated proteins were pelleted by centrifugation (4,500 × g, 20 min, 4°C), washed twice with 50 mL of ice-cold methanol and air-dried. The protein pellet was suspended in 4% SDS buffer (4% SDS, 50 mM triethanolamine pH 7.4, and 150 mM NaCl, 10 mM EDTA). The solubilized protein mixture was diluted to 1% SDS with 1% Brij 97 (in 50 mM triethanolamine pH 7.4, and 150 mM NaCl) and incubated with streptavidin agarose (0.2 ml slurry for 1 mg of protein) for 1 hr at room temperature. The streptavidin beads were washed three times with 10 mL of 1% SDS in PBS buffer. The streptavidin beads were incubated with 1 M $NH_2OH$ (pH 8.0) in 300 μL of 1% SDS PBS buffer for 1 hr at room temperature to elute proteins with only cysteine fatty acylation. The resulting supernatant was concentrated to 20 μL final volume using the Amicon Ultra-0.5 Centrifugal Filter (UFC501008, EMD Millipore). The resulting streptavidin beads were washed three times with 1% SDS in PBS buffer. Both the concentrated supernatant and washed beads were heated in protein loading buffer (2 × final concentration) at 95°C for 10 min and subjected to western blot analyses.

## Confocal microscopy

Cells were seeded in 35 mm glass bottom dishes (MatTek) and transfected with relevant constructs overnight.

To determine RBD binding of K-Ras4a-G12V and -G12V-3KR in Ctrl or SIRT2 KD cells, HEK293T cells were transfected with pCMV5-FLAG-K-Ras4a-G12V or -G12V-3KR and infected with luciferase (Ctrl) shRNA- or human SIRT2 shRNA-carrying lentivirus 12 hr after the transfection. 72 hr later, cells were cultured in FBS-free or complete medium for another 12 hr before being subjected to RBD pull-down as described above.

For live cell imaging, cells were incubated in the Live Cell Imaging Solution (A14291DJ, Thermo Fisher Scientific) and imaged with a Zeiss 880 confocal/multiphoton inverted microscope (Carl Zeiss MicroImaging, Inc., Thornwood, NY) in a humidified metabolic chamber maintained at 37°C and 5% $CO_2$. For time-lapse movies, 60 single section images were recorded at 1 s intervals for 1 min.

For immunofluorescence, cells were rinsed with 1 × PBS twice and fixed with 4% paraformaldehyde (v/v in 1 × PBS) for 15 min. The fixed cells were washed twice with 1 × PBS, permeabilized and blocked with 0.1% Saponin/5% BSA/1 × PBS for 30 min. The cells were then incubated overnight at 4°C in dark with indicated primary antibody at 1/50 - 1/100 dilution (in 0.1% Saponin/5% BSA/ 1 × PBS). Cells were washed with 0.1% Saponin/1 × PBS three times and incubated with Cy3-conjugated goat anti-rabbit IgG (H + L) secondary antibody at 1/1000 dilution (in 0.1% Saponin/5% BSA/ 1 × PBS) at room temperature in the dark for 1 hr. Samples were washed with 0.1% Saponin/ 1 × PBS three times and mounted with Fluoromount-G®(0100–01) from SouthernBiotech before imaging with Zeiss LSM880 inverted confocal microscopy. Images were processed with Fiji software.

For colocalization analyses of GFP-K-Ras4a-WT or −3KR with various intracellular membrane markers, live cell imaging was performed for colocalization with mCherry-Sec61, DsRed-GalT, mCherry-Rab11, DsRed-Rab7 and Lamp1-RFP; immunofluorescence was performed for colocalization with STX6 (1/50 dilution for anti-STX6 antibody) and EEA1 (1/100 dilution for anti-EEA1 antibody).

## Quantitative analyses of colocalization and fluorescence intensity

Fiji software was used for quantification. To quantify the degree of cytoplasmic colocalization, background was subtracted, then the cytoplasm area was selected and quantified for each cell examined. Pearson's correlation coefficient (*Adler and Parmryd, 2010*) was calculated using Fiji plug-in Coloc2 program (http://fiji.sc/Coloc_2) on a single plane between the two indicated fluorescent signals. To quantify fluorescence intensity, background was subtracted and the cytoplasm area or the whole cell was selected for integrated signal intensity quantification. Relative cytoplasm with respect to whole cell fluorescence intensity was presented.

## Soft agar colony formation assay

To assess the effect of Ctrl or Sirt2 KD on K-Ras4a-G12V, -G12V-3KR or H-Ras-G12V-mediated anchorage-independent growth, NIH3T3 cells with stable overexpression of Mock, K-Ras4a-G12V or -G12V-3KR were infected with Ctrl shRNA- or Sirt2 shRNA-carrying lentivirus for 6 hr and cultured in complete medium for another 72 hr before being seeded for soft agar colony formation assay.

To determine the effect of Ctrl or A-Raf KD, NIH3T3 cells with stable overexpression of Mock (pCDH empty vector control), K-Ras4a-G12V or -G12V-3KR were first infected with Ctrl shRNA- or Sirt2 shRNA-carrying lentivirus for 6 hr and then with Ctrl shRNA- or A-Raf shRNAs for another 6 hr. The infected cells were then cultured in complete medium for another 72 hr before being seeded for soft agar colony formation assay.

0.6% base low-melting point agarose (LMP) and 0.3% top LMP were prepared by mixing 1.2% LMP in $H_2O$ and 0.6% LMP in $H_2O$, respectively, with 2 × complete medium in 1:1 (v/v) ratio. 1.5 mL of 0.6% base LMP was added to each well of 6-well plate and allowed to solidify for 30 min at room temperature. Then $5.0 \times 10^3$ cells were resuspended in 0.3% LMP top LMP and plated onto 6-well plate pre-coated with the base LMP. 150 µL of complete medium was added on top of the 0.3% LMP and refreshed every 3 days. After 14 days of culture, colonies were stained with 0.1% crystal violet (m/v in 25% methanol) for 30 min, rinsed with 50% methanol, and counted.

### Cell proliferation assay

NIH3T3 cells with stable overexpression of Mock, K-Ras4a-G12V, or -G12V-3KR were seeded in 12-well plate at a density of $1.5 \times 10^4$ cells/well 24 hr before being infected with luciferase (Ctrl) shRNA- or Sirt2 shRNA-carrying lentivirus for 0 or 5 days. After knocking down Sirt2 for the indicated time, cells were washed with 1 × PBS, fixed with ice-cold methanol for 10 min and then stained with 0.25% crystal violet (m/v, in 25% methanol) for 10 min. The stained cells were washed with running distilled water, air-dried and solubilized in 200–800 µL of 0.5% SDS in 50% ethanol. Absorbance of the resulting solution was measured at 550 nm.

### Transwell migration assay

NIH3T3 cells with stable overexpression of Mock, K-Ras4a-G12V or -G12V-3KR were infected with Ctrl shRNA- or Sirt2 shRNA-carrying lentivirus for 6 hr and cultured in complete medium for another 72 hr. Cells were cultured in serum-free medium for 12 hr before the assay. The assay was performed in 24-well Transwell plate with 8 mm polycarbonate sterile membrane (Corning Incorporated). Cells were plated in the upper chamber (20,000 cells/insert) in 200 µL of serum-free medium. Inserts were then placed in wells containing 600 µL of medium supplemented with 10% FBS. 12 hr later, cells on the upper surface of the filter were detached with a cotton swab and cells on the lower surface of the filters were fixed with ice-cold methanol for 10 min and stained with 0.1% crystal violet for 15 min. The cells were then rinsed with distilled water, photographed and counted. Migration was quantified by counting the migrated cells in ten random microscopic fields.

### Active Ras pull-down and detection

Ras activity was determined using a Ras binding domain of Raf1 (RBD) pull-down assay kit (16117, Thermo Fisher Scientific) by following the manufacturer's instructions. Briefly, to determine RBD binding of K-Ras4a-WT and −3KR in Ctrl or SIRT2 KD cells, HEK293T cells expressing FLAG-K-Ras4a-WT or-3KR were infected with luciferase (Ctrl) shRNA- or human SIRT2 shRNA-carrying lentivirus for 6 hr and cultured in complete medium for another 72 hr. Cells were then serum-starved overnight and treated with 100 ng/mL EGF for 0, 5 and 15 min. At the end of treatment, cells were rinsed with ice-cold 1 × PBS and scraped on ice in lysis buffer containing 25 mM Tris pH 7.2, 150 mM NaCl, 5 mM $MgCl_2$, 1% NP-40 and 5% glycerol and 1 × protease inhibitor cocktail. The samples were collected, vortexed, incubated on ice for 5 min and centrifuged at 16,000 g at 4°C for 15 min to remove cellular debris. Protein concentration was measured by Bradford assay. Equal amounts of lysate (500 µg) were incubated with RBD-coated agarose beads at 4°C for 1 hr. The beads were then washed three times with ice-cold lysis buffer, boiled for 5 min at 95°C, and active Ras was analysed by western blot using Ras-specific antibodies (16117, Thermo Scientific). For comparison to total Ras protein, 1% of total lysates used for pull-down was analysed by immunoblot.

### K-Ras4a interactome by SILAC

Three SILAC experiments were performed to determine K-Ras4a-G12V or -G12V-3KR interacting proteins: (1) NIH3T3 cells stably overexpressing FLAG-K-Ras4a-G12V cultured in DMEM with [$^{12}C_6$, $^{14}N_2$]-L-lysine and [$^{12}C_6$, $^{14}N_4$]-L-arginine as 'light' cells, and NIH3T3 cells stably overexpressing FLAG-K-Ras4a-G12V-3KR cultured in DMEM with [$^{13}C_6$, $^{15}N_2$]-L-lysine and [$^{13}C_6$, $^{15}N_4$]-L-arginine as

'heavy' cells. (2) NIH3T3 cells stably overexpressing FLAG-K-Ras4a-G12V-3KR cultured in DMEM with $[^{12}C_6, {}^{14}N_2]$-L-lysine and $[^{12}C_6, {}^{14}N_4]$-L-arginine as 'light' cells, and NIH3T3 cells stably overexpressing FLAG-K-Ras4a-G12V cultured in DMEM with $[^{13}C_6, {}^{15}N_2]$-L-lysine and $[^{13}C_6, {}^{15}N_4]$-L-arginine as 'heavy' cells. The second group served as the reverse SILAC of the first group. (3) NIH3T3 cells stably overexpressing FLAG-K-Ras4a-G12V and transiently transduced with luciferase (Ctrl) shRNA cultured in DMEM with $[^{13}C_6, {}^{15}N_2]$-L-lysine and $[^{13}C_6, {}^{15}N_4]$-L-arginine as 'heavy' cells, and NIH3T3 cells stably overexpressing FLAG-K-Ras4a-G12V and transiently transduced with mouse Sirt2 shRNA cultured in DMEM with $[^{12}C_6, {}^{14}N_2]$-L-lysine and $[^{12}C_6, {}^{14}N_4]$-L-arginine as 'light' cells.

Cells were collected and lysed in 1% NP-40 lysis buffer containing protease inhibitor cocktail. Protein concentration was quantified by Bradford assay, and 8 mg of total protein from each sample was subjected to FLAG IP to enrich FLAG-K-Ras4a-G12V or -G12V-3KR with its interacting proteins. After washing the FLAG resin five times with IP washing buffer, the resins from 'heavy' and 'light' cells were mixed. Enriched proteins on the resin were eluted with triple FLAG peptide following the manufacturer's protocol. Eluted proteins were precipitated with methanol/chloroform/water (4/1.5/3 vol ratio with the sample volume set as 1), and the protein pellets were washed twice with 1 mL ice-cold methanol. The protein pellets were air dried for 10–15 min, and subjected to disulfide reduction and protein denaturation in 100 µL of buffer containing 6 M urea, 10 mM DTT and 50 mM Tris-HCl pH 8.0 at room temperature for 1 hr. Then iodoacetamide (final concentration 40 mM) was added to alkylate the proteins at room temperature for 1 hr. Subsequently, DTT (final concentration 40 mM) was added to stop alkylation at room temperature for 1 hr. The samples were then diluted seven times with buffer containing 1 mM $CaCl_2$ and 50 mM Tris-HCl pH 8.0 and digested with 2 µg trypsin at 37 °C for 12 hr. Trypsin digestion was quenched with 0.2% trifluoroacetic acid. Then the mixture was desalted using Sep-Pak C18 cartridge following the manufacturer's protocol and subjected to liquid chromatography (LC)-MS/MS analysis.

The lyophilized peptides were reconstituted in 2% acetonitrile (ACN) with 0.5% formic acid (FA) and analyzed by LTQ-Orbitrap Elite mass spectrometer coupled with nanoLC. Reconstituted peptides were injected onto Acclaim PepMap nano Viper C18 trap column (5 µm, 100 µm × 2 cm, Thermo Dionex) for online desalting and then separated on C18 RP nano column (5 µm, 75 µm × 50 cm, Magic C18, Bruker). The flow rate was 0.3 µL/min, and the gradient was 5–38% ACN with 0.1% FA from 0 to 120 min, 38–95% ACN with 0.1% FA from 120 to 127 min, and 95% ACN with 0.1% FA from 127 to 135 min. The Orbitrap Elite was operated in positive ion mode with spray voltage 1.6 kV and source temperature 275 °C. Data-dependent acquisition (DDA) mode was used by one precursor ions MS survey scan from m/z 375 to 1800 at resolution 120,000 using FT mass analyzer, followed by up to 10 MS/MS scans at resolution 15,000 on 10 most intensive peaks. Collision-induced dissociation (CID) parameters were set with isolation width 2.0 m/z and normalized collision energy at 35%. All data were acquired in Xcalibur 2.2 operation software. MS1 and MS2 data were processed using Sequest HT software within the Proteome Discoverer 1.4.1.14 (PD 1.4, Thermo Scientific).

## Detection of lysine fatty acylation on Ras by mass spectrometry (MS)

To detect H-Ras lysine fatty acylation, HEK293T cells were transfected with pCMV5-FLAG-H-Ras for 24 hr and treated with 50 µM Alk14 for another 6 hr. To detect K-Ras4a lysine fatty acylation, HEK293T cells with stable SIRT2 KD were transfected with pCMV5-FLAG-K-Ras4a for 24 hr and treated with or without 50 µM Alk14 for another 6 hr. Cells were collected and lysed in 1% NP-40 lysis buffer with protease inhibitor cocktail. FLAG IP was then performed with 50 mg of total protein lysate to purify FLAG-K-Ras4a or Flag-H-Ras. After washing the FLAG resin three times with IP washing buffer, H-Ras or K-Ras4a was eluted by heating at 95°C for 10 min in buffer containing 1% SDS and 50 mM Tris-HCl pH 8.0. After centrifuging at 15,000 g for 2 min, the supernatant was transferred to a new tube and was treated with 300 mM $NH_2OH$ pH 7.4 at 95°C for 10 min. The Ras protein was then precipitated by methanol/chloroform and processed (disulfide reduction, denaturing, alkylation and neutralization) as described above. The resultant Ras protein was digested with 2 µg of trypsin at 37 °C for 2 hr in a glass vial (to avoid absorption of the fatty acylated peptide by plastics). Then desalting was done using Sep-Pak C18 cartridge following the manufacturer's protocol.

For the LC-MS/MS analysis of the digested peptides, the same settings described for the SILAC experiment were applied except the LC gradient was 5–95% ACN with 0.1% FA from 0 to 140 min. The settings for identifying Alk14 modification in Sequest were: two miscleavages for full trypsin with fixed carbamidomethyl modification of cysteine residue, dynamic modifications of 234.198 Da

(Alk14) on lysine residue, N-terminal acetylation, methionine oxidation and deamidation of asparagine and glutamine residues. The peptide mass tolerance and fragment mass tolerance values were 15 p.p.m. and 0.8 Da, respectively.

## Detection of lysine fatty acylation on endogenous Ras in HCT116 cells

HCT116 cells (parental cells, or cells infected with shCtrl/shSIRT2-carrying lentivirus for 3 days) were cultured with fresh medium containing 50 µM Alk14 for 6 hr. Cells were collected and lysed using the same method described above. Pan-Ras immunoprecipitation was performed using pan-Ras (Y13-259) antibody by following manufacturer's protocol. The lysine fatty acylation on endogenous Ras was detected by on-beads click chemistry and in-gel fluorescence using the same method described above. To directly detect lysine fatty acylation on endogenous Ras by MS, 200 mg of total lysates from HCT116 cells with SIRT2 KD was used for pan-RAS immunoprecipitation, followed by denaturation, alkylation, neutralization, trypsin digestion and LC-MS/MS analysis using the same method described above.

### Statistical analysis

Quantitative imaging data were expressed in box plot as indicated in figure legends. Statistical evaluation of imaging data was done using two-way ANOVA. Other quantitative data were expressed in scatter plots with mean ±SEM (standard error of the mean, shown as error bar) shown. Differences between two groups were examined using unpaired two-tailed Student's t test. The $P$ values were indicated (*$p<0.05$, **$p<0.01$, and ***$p<0.001$). P values < 0.05 were considered statistically significant. No statistical tool was used to pre-determine sample size. No blinding was done, no randomization was used, and no sample was excluded from analysis.

## Acknowledgements

This work is supported in part by a grant (1R01GM121540-01A1) from NIH. HJ is a Howard Hughes Medical Institute International Student Research Fellow. We thank Dr. Sheng Zhang and Dr. Ievgen Motorykin at the Proteomic and MS Facility of Cornell University for help with the SILAC experiments (The Orbitrap Fusion mass spectrometer is supported by NIH SIG 1S10 OD017992-01 grant), the Cornell University Biotechnology Resource Center (BRC) Imaging Facility for help with the confocal microscopy, which is supported by NIH S10RR025502, and Dr. Toren Finkel at NIH for providing the Sirt2 WT and KO MEF cells.

## Additional information

### Funding

| Funder | Grant reference number | Author |
| --- | --- | --- |
| National Institutes of Health | 1R01GM121540-01A1 | Maurine E Linder<br>Hening Lin |
| Howard Hughes Medical Institute | | Hui Jing |

The funders had no role in study design, data collection and interpretation, or the decision to submit the work for publication.

### Author contributions

Hui Jing, Conceptualization, Data curation, Formal analysis, Investigation, Methodology, Writing—original draft, Writing—review and editing; Xiaoyu Zhang, Data curation, Formal analysis, Validation, Investigation, Methodology, Writing—original draft, Writing—review and editing; Stephanie A Wisner, Data curation, Writing—review and editing; Xiao Chen, Validation, Writing—review and editing; Nicole A Spiegelman, Resources, Writing—review and editing; Maurine E Linder, Resources, Funding acquisition, Writing—review and editing; Hening Lin, Conceptualization, Resources, Supervision, Funding acquisition, Writing—original draft, Project administration, Writing—review and editing

## Author ORCIDs

Xiaoyu Zhang (iD) http://orcid.org/0000-0002-0951-9664

Hening Lin (iD) http://orcid.org/0000-0002-0255-2701

## Decision letter and Author response

Decision letter https://doi.org/10.7554/eLife.32436.025

Author response https://doi.org/10.7554/eLife.32436.026

## Additional files

### Supplementary files

• Transparent reporting form

DOI: https://doi.org/10.7554/eLife.32436.023

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
