## [Decision Letter]

Thank you for submitting your article "SIRT2 and lysine fatty acylation regulate the oncogenic activity of K-Ras4a" for consideration by *eLife*. Your article has been favorably evaluated by Michael Marletta (Senior Editor) and three reviewers, one of whom is a member of our Board of Reviewing Editors. The reviewers have opted to remain anonymous.

The reviewers have discussed the reviews with one another and the Reviewing Editor has drafted this decision to help you prepare a revised submission.

Summary:

This study reports the discovery and characterization of Lys-fatty-acylation on Ras small GTPases and their regulation by SIRT2. Using metabolic labeling and click chemistry detection, the authors demonstrate that H-Ras, N-Ras and K-Ras4a are hydroxylamine-resistant and Lys-fatty-acylated. The authors then show that K-Ras4a Lys-fatty-acylation is primarily regulated by SIRT2 deacylase activity by in vitro studies and shRNA-KD. For K-Ras4a, the authors attempt to address endogenous levels of Lys-fatty-acylation by MS/MS experiments and suggests the modification may occur on K182/184/185 and, intriguingly, is coupled to prenylation of C186. Studies with K-R mutants suggest that Lys-fatty-acylation regulates K-Ras4a distribution between PM and endosomes, A-Raf G12V-mediated transformation of NIH3T3 cells and activation of A-Raf effector signaling, which may also be regulated by SIRT2. Collectively, the authors provide intriguing in vitro and cellular evidence that 3/4 Ras-GTPases are Lys-fatty-acylated and that SIRT2 regulates K-Ras4a intracellular localization and downstream signaling functions, particular in cellular transformation. These studies provide a potential link between fatty acid and NAD metabolism in cancer and should motivate additional studies in normal physiology and disease.

Essential revisions:

The general findings are, in principle, of interest to the readership of *eLife*. However, the following points should first be addressed:

1) The two different assays for localization of K-Ras4a versus K-Ras4a-G12V raise questions about the actual role of fatty acylation in regulating these two proteins. The modest effects in Figure 4 are based on analysis of punctate signals. However, in Figure 6, there is not much intracellular punctate localization for K-Ras4a-G12V, and A-Raf looks colocalized only with membrane-bound K-Ras4a-G12V. These results seem to suggest that there is a difference in fatty acylation levels between K-Ras4a and K-Ras4a-G12V. The authors should account for these discrepancies and provide information on the fatty acylation levels of K-Ras4a versus K-Ras4a-G12V.

2) The authors have primarily used K-R mutants as "loss of Lys-fatty-acylation" alleles. However, the multiple Arg substitutions have the potential to introduce alternative modes of membrane binding independent of fatty-acylation. Indeed, in several figures (Figure 5, Figure 5—figure supplement 1, Figure 6—figure supplement 1), the effects of the 3KR substitution seemed largely independent of SIRT2. It would be important for the authors to include activity and location data on Lys -> Ala mutants, especially for K-Ras4a, in order to support the conclusions.

3) The effects of SIRT2 knockdown in Figure 4 and Figure 5, as well as in Figure 2 are small and thus not entirely convincing. This may be due to an incomplete knockdown, for example as seen in Figure 2. More significant effects that may be obtainable by knocking out SIRT2 would make a more convincing case for the role of SIRT2 in regulating K-Ras4a.

---

## [Author Response]

Essential revisions:The general findings are, in principle, of interest to the readership of eLife. However, the following points should first be addressed:1) The two different assays for localization of K-Ras4a versus K-Ras4a-G12V raise questions about the actual role of fatty acylation in regulating these two proteins. The modest effects in Figure 4 are based on analysis of punctate signals. However, in Figure 6, there is not much intracellular punctate localization for K-Ras4a-G12V, and A-Raf looks colocalized only with membrane-bound K-Ras4a-G12V. These results seem to suggest that there is a difference in fatty acylation levels between K-Ras4a and K-Ras4a-G12V. The authors should account for these discrepancies and provide information on the fatty acylation levels of K-Ras4a versus K-Ras4a-G12V.

We agree with the reviewers that K-Ras4a-G12V (Figure 6) exhibits weaker intracellular punctate localization than K-Ras4a (Figure. 4A) (*P* < 0.01 by two-way ANOVA analysis). However, K-Ras4a and K-Ras4a-G12V possess comparable total and lysine fatty acylation levels (Figure 4—figure supplement 1), suggesting that the difference in their subcellular localization is not due to different levels of fatty acylation. It is possible that the activation state of K-Ras4a influences its dynamics on plasma membrane and endomembranes, the mechanism for which remains to be elucidated in the future. We have added discussion on this in the revised manuscript (Discussion, fifth paragraph).

We apologize for not clearly showing the intracellular colocalization of K-Ras4a-G12V and A-Raf. We have added a Merge 2 column in the revised Figure 6 to show magnified regions in Merge 1 and to better demonstrate the endomembrane colocalization of K-Ras4a-G12V and A-Raf. The figure legend for Figure 6 has been revised accordingly.

2) The authors have primarily used K-R mutants as "loss of Lys-fatty-acylation" alleles. However, the multiple Arg substitutions have the potential to introduce alternative modes of membrane binding independent of fatty-acylation. Indeed, in several figures (Figure 5, Figure 5—figure supplement 1, Figure 6—figure supplement 1), the effects of the 3KR substitution seemed largely independent of SIRT2. It would be important for the authors to include activity and location data on Lys -> Ala mutants, especially for K-Ras4a, in order to support the conclusions.

3KR mutation may not be solely due to lack of lysine fatty acylation. Therefore, we think it is critical to utilize K-Ras4a-3KR mutant in combination with SIRT2 KD to study the function of K-Ras4a lysine fatty acylation. If a biological effect is due to lysine fatty acylation, SIRT2 KD should significantly affect K-Ras4a WT but not 3KR mutant in the most ideal case. This is indeed what we observed for the endomembrane localization (Figure 4), transforming activity (Figure 5) and A-Raf binding (Figure 6).

Recently, Tsai et al.^1^ and Zhao et al.^2^ showed that the lysine to uncharged glutamine mutation (3KQ) decreased transforming activity of K-Ras4a-G12V, suggesting that positive charge is favorable for K-Ras4a activity. Based on these studies, the lysine to alanine mutation (KA), which does not preserve the positive charge, may add more complexity to the membrane binding modes of K-Ras4a than the KR mutation. Any effect caused by the KA mutation may be due to lack of lysine fatty acylation, lack of positive charge, or alternative modes of membrane binding independent of fatty acylation, which will make the interpretation of the results more complicated and therefore will not further support our conclusions. In contrast, with the KR mutant, we only need to consider whether the effect is due to lysine fatty acylation or alternative modes of membrane binding independent of fatty acylation. In other words, we think the KR mutant is the closest to the WT without lysine fatty acylation (i.e. with SIRT2 present) and thus the most logical to use for our purpose.

3) The effects of SIRT2 knockdown in Figure 4 and Figure 5, as well as in Figure 2 are small and thus not entirely convincing. This may be due to an incomplete knockdown, for example as seen in Figure 2. More significant effects that may be obtainable by knocking out SIRT2 would make a more convincing case for the role of SIRT2 in regulating K-Ras4a.

As the reviewers suggested, we indeed tried to compare the lysine fatty acylation and subcellular localization of K-Ras4a in *Sirt2* wildtype (WT) and knockout (KO) mouse embryonic fibroblast (MEF) cells. Lysine fatty acylation level for K-Ras4a-WT in the KO cells was 1.56-fold of that in the WT cells, whereas lysine fatty acylation of K-Ras4a-3KR was similar in WT and KO cells (Figure 3—figure supplement 1, subsection “Mapping the fatty acylated lysine residues regulated by SIRT2”, last paragraph in the revised manuscript). In *Sirt2* WT MEF cells, K-Ras4a-3KR displayed slightly but significantly more intracellular punctate localization than K-Ras4a-WT. K-Ras4a-WT showed significantly less intracellular punctate localization in KO cells than in WT cells, whereas K-Ras4a-3KR exhibited comparable level of punctate localization in WT and KO cells (Figure 4—figure supplement 1, subsection “Lysine fatty acylation regulates subcellular localization of K-Ras4a”, first paragraph). Collectively, these effects of *Sirt2* KO in MEF cells are consistent with that of SIRT2 KD in HEK293T cells, which further strengthens our conclusion that SIRT2-mediated lysine defatty-acylation regulates K-Ras4a intracellular localization. However, we did not observe more significant effects of *Sirt2* KO in MEF cells than that of SIRT2 KD in HEK293T cells, which is likely because we had over 95% of knockdown efficiency (based on quantification of SIRT2 signal intensity in Figure 2) with shSIRT2-#2 in HEK293T cells.

We have also attempted to examine the transforming activities of K-Ras4a-G12V and -G12V-3KR in *Sirt2* WT and KO MEF cells. However, introduction of both K-Ras4a-G12V and -G12V-3KR led to oncogene-induced senescence in both WT and KO cells. Considering that *Sirt2* KO shows similar effects on K-Ras4a lysine fatty acylation and subcellular localization to SIRT2 KD, we believe that our current data obtained from 3T3 cells with near 95% SIRT2 KD efficiency (Figure 6) is significant enough to support our conclusion.

**References**

1) Tsai, F.D., Lopes, M.S., Zhou, M., Court, H., Ponce, O., Fiordalisi, J.J., Gierut, J.J., Cox, A.D., Haigis, K.M. & Philips, M.R. K-Ras4A splice variant is widely expressed in cancer and uses a hybrid membrane-targeting motif. Proc Natl Acad Sci U S A (2015).

2) Zhao, H., Liu, P., Zhang, R., Wu, M., Li, D., Zhao, X., Zhang, C., Jiao, B., Chen, B., Chen, Z. & Ren, R. Roles of palmitoylation and the KIKK membrane-targeting motif in leukemogenesis by oncogenic KRAS4A. J Hematol Oncol 8, 132 (2015).